# From single-sequences to evolutionary trajectories: protein language models capture the evolutionary potential of SARS-CoV-2

Kieran D. Lamb [1,2], Joseph Hughes [1], Spyros Lytras [1,3], Francesca Young[1], Orges Koci[1,4], James C. Herzig[5], Simon C. Lovell[5], Joe Grove [1], Ke Yuan [2,6,7] ✉ & David L. Robertson [1] ✉

Protein language models (PLMs) capture features of protein three-dimensional structure from amino acid sequences alone, without requiring multiple sequence alignments (MSA). The concepts of grammar and semantics from natural language have been suggested to have the potential to capture functional properties of proteins. Here, we investigate how these representations enable assessment of variation due to mutation. Applied to the SARS-CoV-2 spike protein via in silico deep mutational scanning (DMS), the PLM ESM-2 captures evolutionary constraints directly from sequence context, recapitulating what normally requires MSA data. Unlike other state-of-the-art methods which require protein structures or multiple sequences for training, we show what can be accomplished using an unmodified pretrained PLM. Applied to SARS-CoV-2 variants across the pandemic, we demonstrate that ESM-2 representations encode the evolutionary history between variants, as well as the distinct nature of variants of concern upon their emergence, associated with shifts in receptor binding and antigenicity. ESM-2 likelihoods can also identify epistatic interactions among sites in the protein. Our results here affirm that PLMs like ESM-2 are broadly useful for variant-effect prediction, including unobserved changes, and can be applied to understand novel viral pathogens with the potential to be applied to any protein sequence, pathogen or otherwise.

The conventional study of protein sequence variation requires the alignment of amino acid sequence data. Alignments help identify where variation accumulates within sequences, revealing the evolutionary constraints that dictate observed variation. In human genome studies, impactful missense mutations, non-synonymous substitutions resulting in amino acid changes, are primarily linked to disease, while in viruses, amino acid changes are more often linked to adaptive evolution[1,2]. Alternatively, these changes (and in fact the majority of

[1]MRC-University of Glasgow Centre for Virus Research, School of Infection and Immunity, Glasgow, UK. [2]School of Computing Science, University of Glasgow, Glasgow, UK. [3]Division of Systems Virology, Department of Microbiology and Immunology, Institute of Medical Science, University of Tokyo, Tokyo, Japan. [4]European Molecular Biology Laboratory- European Bioinformatics Institute, Hinxton, UK. [5]School of Biological Sciences, University of Manchester, Manchester, UK. [6]School of Cancer Sciences, University of Glasgow, Glasgow, UK. [7]Cancer Research UK Scotland Institute, Glasgow, UK. ✉e-mail: ke.yuan@glasgow.ac.uk; david.l.robertson@glasgow.ac.uk

**Fig. 1 | Schematic methodology summary.** Deep mutational scanning involves taking a sequence and mutating every position to each of the possible alternative amino acids. These are then passed to the PLM to produce embeddings and logits which can be used for downstream tasks or to produce the metrics relative grammaticality and semantic score. Identifying epistatic interactions involves reverting the mutations from a SARS-CoV-2 variant (here BA.1) and measuring the effect this has on the other likelihoods. Embeddings and log-likelihoods (logits) can be used for surveillance by producing metrics over timescales for newly emergent protein sequences of concern, or by looking at evolutionary trajectories such as with evo-velocity.

possible missense mutations) are deleterious resulting in, e.g., misfolding and a decrease in fitness or are "neutral" and have little to no consequence.

To date approaches for assessing the impact of mutations are premised on the availability of comparative genome sequence data to assess features like the relative proportions of non-synonymous to synonymous substitutions (dN/dS) or evolutionary conservation at individual sites. Similarly, genome sequence-based surveillance methods usually require aligned real-time sequence data to assess the relative growth rate of a pathogen lineage[3,4]. Experiments on novel variants can be initiated when first detected, but they take considerable time to complete.

Natural language processing methods can be applied to biological sequences like DNA and proteins[5–7]. Biological sequences are physical molecules represented by arbitrary characters. This is convenient since these characters, representing nucleotides or amino acids, can be processed in the same way as words in natural language. Hie. et al.[6] show how, much like natural language, biological sequence context can be used to create informative embeddings that reflect complex properties of the sequence. Specifically, concepts like grammar (the structural rules of a language) and semantics (the meaning of the language) can purportedly be applied to changes in protein amino acid sequences to capture 'fitness' and mutation effects, respectively[6]. The 'grammaticality' of a sequence is captured by its probability, indicating how well it conforms to the rules of protein formation and composition. The semantic difference between protein sequences can be measured by computing distances, a 'semantic score', between sequences within the embedding space.

Evolutionary scale model 2 (ESM-2)[5,7] is a protein language model (PLM) trained on approximately 65 million unique protein sequences[5,7]. Here, we demonstrate that ESM-2 can infer fundamental properties that classically require a multiple sequence alignment. Applied to SARS-CoV-2 we show that the grammaticality and semantic score metrics reveal characteristic properties of the SARS-CoV-2 spike protein sequence (Fig. 1). We demonstrate these metrics have a broader meaning when computed using ESM-2 compared with previous models, likely due to differences in training data[6], before assessing how to use these metrics for variant and sequence horizon scanning. Crucially, these metrics, produced from a single protein sequence using pre-trained models, bypass the need for alignment data. We make a case for their future application in characterising and assessing the constraints on proteins from emerging pathogens, before assessing how to use these metrics for variant and sequence horizon scanning.

## Results and discussion
### ESM-2 captures evolutionary potential
We first assess how well ESM-2 captures constraint on sequence variation from protein structure. Specifically, does the model assign appropriate values for relative grammaticality (the log-likelihood ratio between the mutation and the reference amino acid) and semantic score (the distance between sequence embeddings, e.g., mutant and wild-type) at different sites in the protein sequence. To do this analysis, we utilised an in silico deep mutational scanning (DMS) approach[8] using embeddings for every possible single amino acid replacement in the SARS-CoV-2 spike protein. In vitro deep mutational scans involve

experimentally substituting each site in a protein with every other amino acid and measuring the substitutions' effect on phenotype. We do this computationally by embedding each of these mutated sequences using ESM-2 and then calculating grammaticality and semantic scores for each sequence.

The ESM-2 scores correlate well with the structural subunits of the spike (Fig. 2). Using our DMS approach to produce relative grammaticality and semantic scores, we observe that the protein has two main regions that broadly correspond to the two protein subunits S1 and S2 (Fig. 2B, C). Relative grammaticalities decrease when substitutions are introduced in the S2 indicating that the region is less tolerant to change compared to S1. We observe a statistically significant difference between the mean relative grammaticalities of the S1 ($n = 685$ positions) (Supplementary Fig. 1) and S2 ($n = 588$ positions) subunits ($p = 2.60\text{e-}160$, two sided Mann–Whitney U test). Spike is a trimer with the core of the S2 subunit being formed by three closely interacting alpha helices from each monomer. As such, changes in this region could disrupt the formation of a stable and functional spike trimer by disrupting the inter-monomer interactions. On the other hand, in the S1 subunit, mutations in the N-terminal Domain (NTD) and receptor-binding domain (RBD) have higher relative grammaticalities. Despite being important to protein function, these regions need to be sufficiently flexible to facilitate receptor binding interactions and accommodate mutations to evade host immunity. This mirrors the accumulation of sequence variation in alignments. Specifically, using entropy, a measure of site-specific variation computed from multiple sequence alignments, there is significantly higher entropy observed at sites in S1 compared to S2 (Fig. 2A), with each region found to be significantly different ($p = 1.121\text{e-}10$, two sided Mann–Whitney U test) (Supplementary Fig. 1). Thus, ESM-2 has correctly identified S1 as a more variable region than S2 (an analysis based on permutations of a single sequence) and indicates that ESM-2 has learnt this property about spike. Available full spike experimental DMS results from Dadonaite et al.[9] show binding is the only measure with a significant difference between subunits ($p = 1.70\text{e-}06$, two sided Mann–Whitney U test), which might be expected given that S1 contains the spike binding domain.

The semantic rank also increases in the NTD region of the S1 (Fig. 2A, B), indicating that changes here may produce large structural changes and are more likely to be accommodated due to higher relative grammaticalities across the region; presumably reflecting properties of the NTD, i.e., it is a region on the surface of the protein and contains many antigenic epitopes[10]. These results are consistent with prior results that show PLMs like ESM-2 can be used as an effective predictor of variant effect[11]. If PLMs can model regional constraints, they may also be able to identify epistatic interactions that are linked with protein structural constraints.

Our results show ESM-2 models spike protein constraints, mapping low relative grammaticalities to the S2 and high relative grammaticalities to the S1, which is heavily targeted by host antibodies (NTD, RBD)[10,12–15] and predominantly on the protein surface (Fig. 2). Hie et al.[6] use the semantic score as a proxy for antigenic change, however, this is not necessarily relevant for protein sites that are not involved in antibody binding. Since ESM-2 is trained on a wide diversity of proteins, semantic scores produced by this model represent shifts in structure/function rather than specifically antigenic change. Many high semantically ranked changes occur in the NTD (Fig. 2B) which contains a "supersite" with several epitopes targeted by neutralising antibodies[10]. The NTDs low relative grammaticality and high semantic score suggests it allows large structural changes to alter its antigenic properties without greatly affecting function.

## ESM-2 identifies epistatic interaction sites

A consequence of protein structural constraint is that compensatory mutations are often required before a change can be accommodated.

Such epistatic effects can be due to direct or indirect mechanisms linked to residues in close contact in the protein structure conformation and/or protein stability[16,17]. Here, we present an approach for assessing epistasis, where we find putative epistatic interactions among sites by identifying where changes in the amino acid at one site cause significant changes in amino acid likelihoods at another. Language models compute likelihoods for each amino acid in a sequence based on the context of the rest of the sequence. This means that the likelihoods depend on the other amino acids in the sequence and will change if other positions mutate. Positions that are unaffected by the mutated site would be expected to change minimally, whereas those affected may change their probabilities, i.e., lower or higher depending on whether an epistatic interaction is antagonistic or synergistic, respectively. A method by Zhang et al.[18] uses a similar approach to predict contact maps, however, here we want to predict mutation specific effects instead of site specific contacting residues.

The BA.1 Omicron, variant of concern (VOC) is defined by over 30 unique substitutions in its spike protein sequence. By reverting these defining mutations to the Wuhan-Hu-1 reference sequence amino acid, we measure the effect this reversion has on the probabilities of all other BA.1 amino acids computed by the model. To illustrate these results we highlight the interactions of three key substitutions E484A, D614G and N969K[19–22]; selected due to their effects on antigenicity, conformational changes and presence in the S2 domain. For ease of interpretation, we inversed these values so that positive scores can be interpreted as a synergistic and negative scores antagonistic when the mutation is acquired. To illustrate these results we highlight the key substitutions E484A, D614G and N969K[19–22]; selected due to their effects on antigenicity, conformational changes and presence in the S2 domain.

The E484A substitution in the RBD of spike is involved in modulating binding to the ACE-2 receptor and is a good example of a substitution with localised effect (Fig. 3A). The VOCs Beta and Gamma contained the E484K substitution and this contributed to escape from several RBD-neutralising antibodies[23]. Interestingly, A, D, G and K substitutions at position 484 all confer resistance to human convalescent sera[14,24]. The mutation primarily changes amino acid likelihoods within the RBD (Fig. 3A).

Interestingly, RBD positions 482, 485, 486, 488 and 500 have the largest absolute changes in probability following the E484A mutation's occurrence: 485 and 482 decrease their likelihood of changing indicating incompatibility with the E484A. Positions 486, 488, 500 increase their likelihoods of changing following the E484A mutation, indicating that this mutation is synergistic with these positions, possibly improving the structure or at least minimising disruption. Omicron mutation Q498R also increases its likelihood in response to E484A. E484A shows decreased immune evasiveness relative to E484K, however interactions between N501Y and Q498R mutations recover this[20]. While 501Y is not identified, position 500 increases in likelihood suggesting synergistic interactions with this region.

Our results show S1 mutations like E484A have shorter distances to their affected positions in sequence space and in three-dimensional space (Supplementary Fig. 3) likely due to it being the subunit comprising most of the accessible surfaces of the protein. Changes more 'central' to the protein structure like those in the S2 domain have a greater chance to affect more amino acids due to being surrounded by other sites. This can produce more impactful changes that have knock-on effects at distant locations.

The D614G substitution emerged early in the pandemic[25], is now present in all circulating lineages, and is an example of a substitution with broad effect (Fig. 3A). Specifically, D614G has two main functional consequences: (1) to remove a hydrogen bond between 614 and 859 on the adjacent spike monomer, and (2) at 647 to contribute to the structure of the C-terminal domain. 614 G increases the stability of the spike protein[19] as well as the infectivity of the virus at the cost of

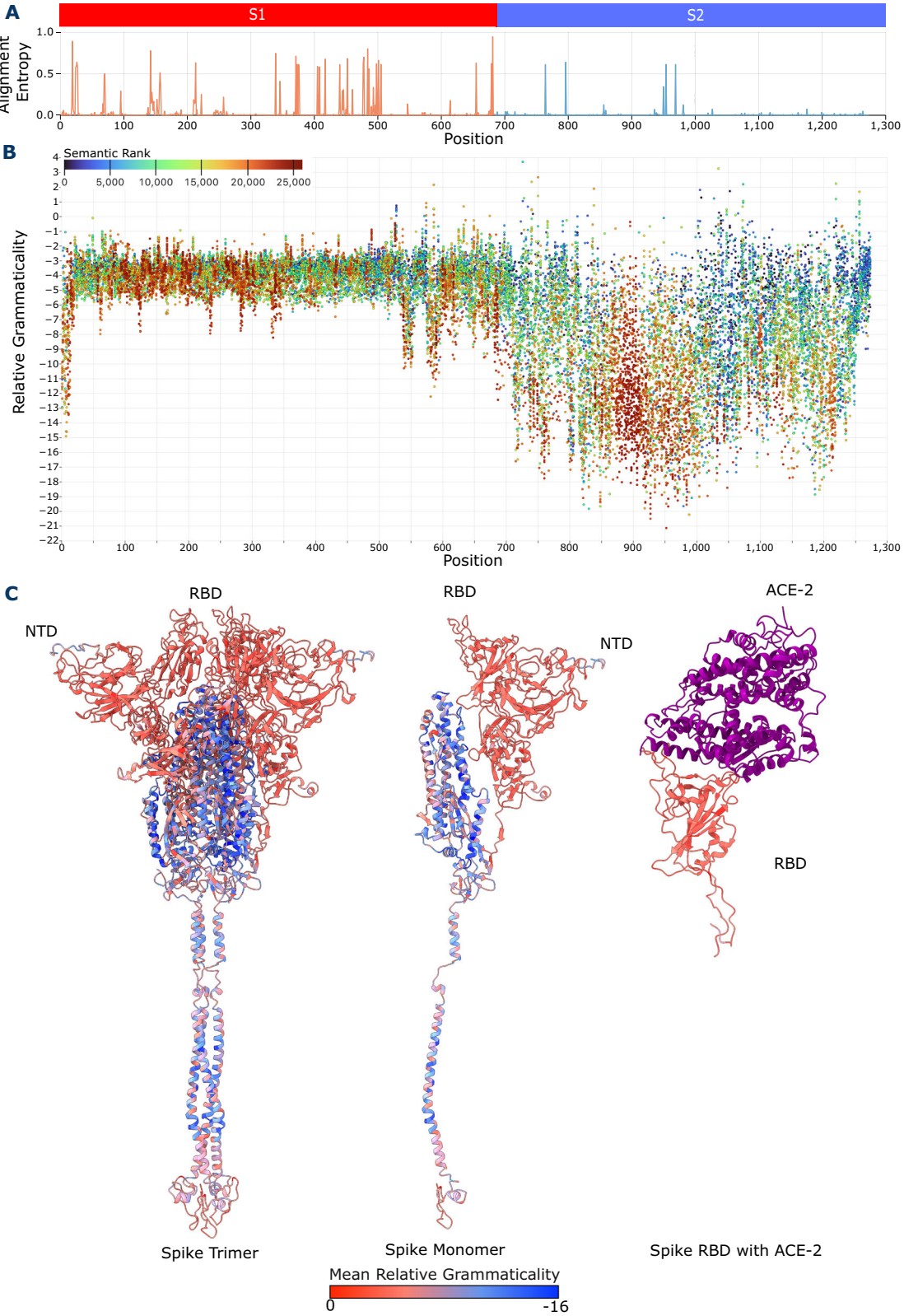

**Fig. 2 | ESM-2 identifies where variation accumulates in Spike. A** Graph of the amino acid sequence alignment entropy at each position in the SARS-CoV-2 spike protein. S1 contains the majority of the sites with high entropy, while S2 contains only a few. **B** Scatter plot of spike protein DMS. Relative grammaticality is shown on the y-axis, with the amino acid position on the x-axis. Points are coloured on the semantic rank of each change. **C** Average relative grammaticality at each position on the spike protein plotted on three structures[66], the full Spike protein, the spike monomer, and the spike receptor binding domain (RBD) bound to the ACE-2 receptor in purple.

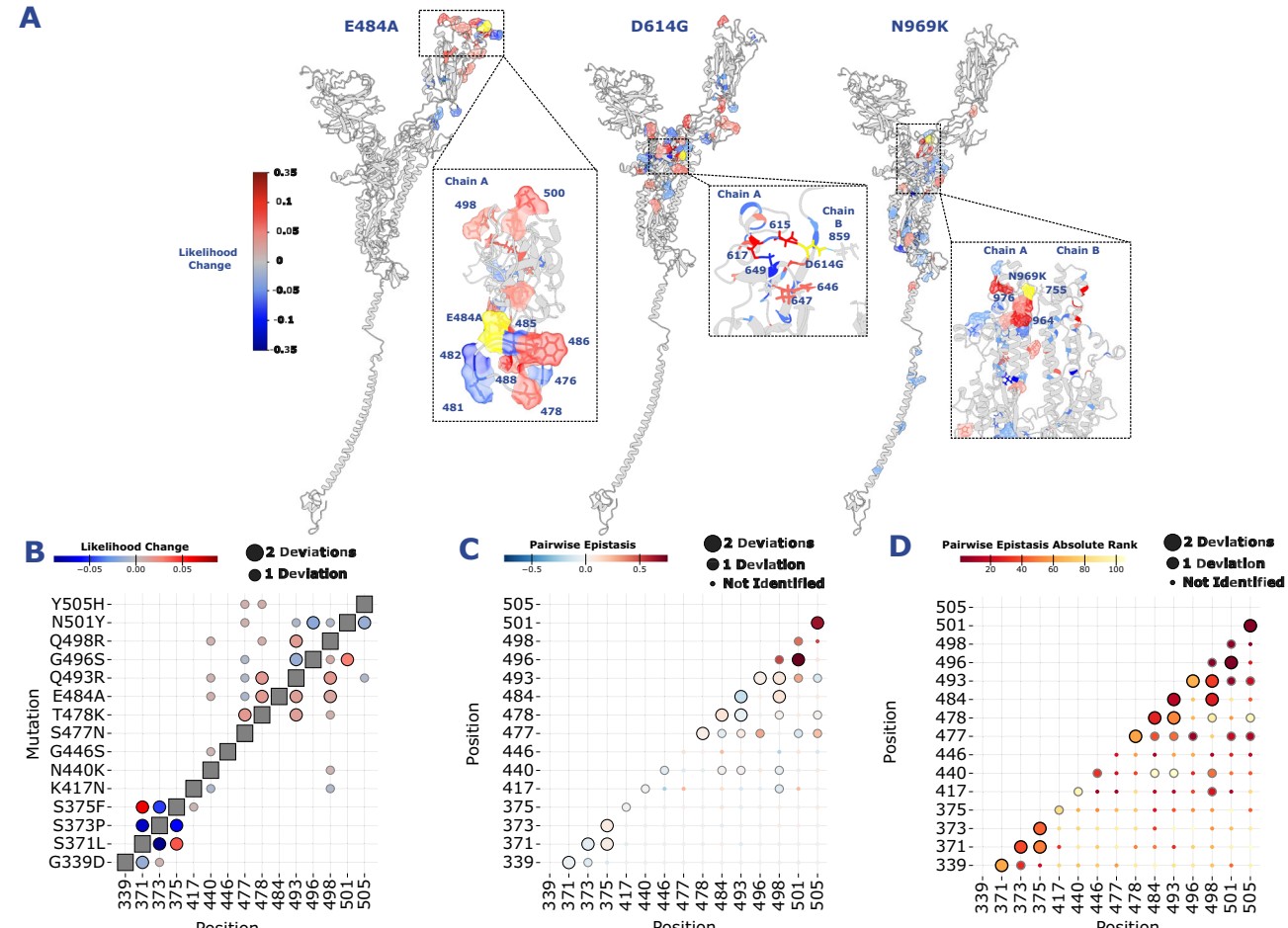

**Fig. 3 | ESM-2 identifies putative epistatic interactions. A** Monomeric structures showing the changes in probabilities for three example substitutions in SARS-CoV-2's spike protein: E484A, D614G and N969K. This is the negative of the reversion likelihood. The mutation site is coloured yellow, red sites increase in probability while blue sites decrease. Mutation probabilities were only shown if they were outside two standard deviations of the mean change from all BA.1 reversions. **B** Heatmap of BA.1 RBD mutations and the change in likelihoods at other BA.1 mutation sites. Larger circles indicate the sites were two standard deviations away from the average change, smaller circles indicate 1 standard deviation away.

**C** Experimental epistasis data from Moulana et al.[28] showing the pairwise effect of mutations on ACE-2 binding affinity. Positive values indicate a pairwise increase in ACE-2 binding affinity. Larger circles indicate the sites were identified by ESM using a two standard deviation delta from the average change, smaller circles indicate 1 standard deviation away. The smallest points were not identified by ESM.
**D** Experimental epistasis data coloured on the ranking of the absolute epistatic effect values. This allows large negative values to be counted as high ranking and vice versa.

increased susceptibility to neutralising antibodies[19,26]. Its reversion affected 46 other positions (Supplementary Fig. 4), nine of which are in the RBD, with a further 30 in S1 and seven in S2. While 859 and 647 were not affected positions, 615, 617 and 649 did change their likelihoods. 649 is an internal amino acid located behind 614 while 615 is in direct contact with the site and 617 is slightly downstream. 617 and 615 both increase their likelihoods, indicating that the D614G acquisition was synergistic with those sites. 649 decrease its likelihood by 3.7% on reversion suggesting an antagonistic effect with 614 G.

The N969K reversion affected 46 positions on the protein (Fig. 3, Supplementary Figs. 3 and 4) and has been linked to the change in Omicron entry route[21]. N969K forms electrostatic contacts with Q755 on the adjacent protomer in the pre-fusion state[27], as well as interacting with and displacing the HR2 backbone in the post-fusion spike structure[22]. It has long-range interactions both in linear sequence and in the 3D protein structure context (Supplementary Fig. 3A) and also has the greatest number of affected sites compared with the other BA.1 mutations. Position 976 had a likelihood increase of 21% while position 964 had a 15.9% increase, indicating a strong synergistic effect with the N969K substitution. N969K is positioned in the middle of a loop connecting two alpha helices, with 976 and 964 positioned at the

end of the first helix and the start of the second. The large likelihood shift indicates strong synergy with the N969K mutation and could be as yet undiscovered epistatic positions. Both D614G and N969K have interacting residues on other monomers, however ESM was only given a single monomer sequence. The lack of other monomers in the embedding may prevent the detection of these interactions.

To further validate that ESM-2 was detecting epistatic effects with other sites, we compared the RBD positions identified by our BA.1 reversions with experimental data from Moulana et al.[28], which characterised the epistatic interactions involved in BA.1's increased ACE-2 binding affinity. We used two cutoffs, a relaxed threshold using a single standard deviation and a stringent two standard deviation threshold. We found that the identified RBD mutational interactions appear to be enriched with high ranked epistatic changes (Fig. 3B–D) in both thresholds. Using a hypergeometric test, the stringent threshold was found to be 6.43 fold over enriched ($n = 267$, p-value 1.46e-10) with RBD-RBD BA.1 mutation interactions. This was also true for strong RBD-RBD mutation epistatic interactions ($n = 267$, an over enrichment of 8.21 fold, p-value 0.00562) as well as for all BA.1 mutation-mutation interactions ($n = 653$, 1.6 fold, p-value 0.01687). Strong epistatic interactions were defined as 1 standard deviation from the mean

absolute experimental epistatic interaction. These remained significant after applying a Benjamini-Hochberg multiple test correction.

The mutation sets with the largest pairwise effects 496–501 and 501–505 are identified in the most stringent cutoff, with the next biggest impacting pairs 496–498 and 498–501 identified in the less stringent set (Fig. 3B–D). While all mutational pairs measure some difference, the ESM-identified changes appear to be predominantly impactful changes. Interestingly, the strongest epistatic effects that ESM-2 identifies between positions 371, 373 and 375 (Fig. 3B) are only found to be weakly involved in ACE-2 binding epistatic effects (Fig. 3C, D). However, recent experimental data from Furnon et al.[29] has shown that these sites as well as 339 appear to be critical determinants of BA.1's enhanced replication in nasal epithelial cells, a major phenotypic feature of Omicron SARS-CoV-2 viruses. These mutations are also highlighted by Moulana et al.[28] due to their significant interactions with each other, despite their smaller effect on ACE-2 binding. As our measure of epistasis is not directly linked to ACE-2 binding and is simply a change in likelihood of the amino acid at the site, it is probable that ESM-2 is picking up on multiple epistatic effects rather than just binding affinity. The lack of direct relatedness to binding affinity may also explain why predicted effects in our analysis (Fig. 3B) do not always match the direction (positive or negative) to the results reported by Moulana et al.[28]. Several factors are at play in determining the suitability of mutations within the RBD including immune escape potential as well as binding affinity. Encouragingly, we also identify a larger set of the Moulana et al. epistatic interactions compared to an epistatic model[30] utilising the pandemic sequence data to identify interactions (Supplementary Fig. 12).

We also tested the sensitivity and specificity of the model in comparison to the mutual information based approach from Innocenti et al.[30] and found that when looking within the experimental data for epistatic sites, ESM-2 (stringent 2 standard deviation cut-off) had an equivalent sensitivity (0.120 vs 0.114) and improved specificity (0.842 vs 0.777). This was calculated using the strongly epistatic set of RBD-RBD mutation interactions. While the specificity is good for both approaches, suggesting both methods work well for ruling out false positive epistatic interactions (or in this instance, weak interactions), the low sensitivity suggests that the task of identifying true/strong interactions remains challenging. Considering ESM-2s lack of SARS-CoV-2 pandemic sequence data in training, future work could be directed towards leveraging pandemic data to improve the predictive performance, although the lack of extensive validation/training experimental pairwise datasets may prove to be a bottleneck here. Also, it should be again noted that ESM-2 is not trained explicitly to predict effects on ACE-2 binding, meaning the epistatic signal may be from another biologically meaningful measure, like the 371, 373 and 375 mutations.

In our epistasis analysis, position 330 changes its likelihood in the mutations mentioned above (E484A, D614G and N969K) and is also associated with several BA.1 mutations. This prompted an investigation into whether certain sites were more prone to fluctuations than others, and whether these sites were of functional importance. We identified positions that significantly changed their likelihoods in each DMS mutant (putative 'critical' sites), before filtering for sites that occur across many different mutations and investigating their functional relevance to the protein. We also investigated which sites when mutated significantly affect a larger number of other sites (putative 'impactful' sites).

The critical sites in the NTD and RBD are predominantly prolines and cysteines, while elsewhere there is a larger range of amino acid types, seven of which contain aromatic side chains (F, Y and W) (Fig. 4A, B). These amino acids have important structural features. Proline is a rigid amino acid due to its side-chain which reduces its flexibility[31,32] and prevents it from forming stable α-helices[31]. Prolines are often found where sharp turns are necessary for structure. Cysteine

thiol side-chain allows them to form covalent di-sulphide bridges between adjacent cysteines[33]. Cysteines are well conserved throughout proteins[33], and all of the identified cysteines were involved in di-sulphide bridges (Fig. 4A, B).

The impactful sites are also heavily enriched with cysteines throughout the protein, affirming their critical importance to protein structure (Fig. 4C, D). Two positions within the furin cleavage site (FCS) 682 and 685 are also highlighted. Both are arginine, a necessary polybasic component of the cleavage site that allows cleavage by furin[34]. The FCS is an important feature of SARS-CoV-2 that enhances transmission[35]. Critical and impactful sites play key structural and functional roles in the protein, with clear identification of structurally important amino acids such as cysteines and functionally relevant positions within the FCS.

## ESM-2 provides orthogonal metrics for assessing mutational impact

An important question is whether the grammaticality and semantic score represent a different and useful measure of change compared with existing experimental, computational or evolutionary measures of mutation impact. We also wanted to determine whether these scores equate to fitness and antigenicity[6] when computed using a general PLM like ESM-2. Extensive high-throughput experimental investigations have revealed various aspects of spike protein biology. We sought to correlate these measurements with PLM metrics, to better understand what characteristics are captured by the model. We calculated spearman's rank correlations between the embedding metrics and experimental and computational metrics. We then used the full embeddings and logits as well as score combinations to fit linear and support vector regressions (SVR) to each metric and again scored using a spearman's rank correlation.

We benchmarked our results against two other computational scores: EVEscape from Thadani et al.[36] and mutability scores from Rodriguez-Rivas et al.[37]. EVEscape is a variational autoencoder based on EVE[38] that was trained using pre-pandemic spike protein sequences. These prior sequences are given to EVEscape as a one-hot encoded alignment, which the model uses to learn distributions and dependencies across sites in the protein. The variational autoencoder produces a mutation score that when compared with the score of the reference sequence approximates the relative fitness of the mutant, termed EVEscape Fitness. This is combined with a structure derived contact accessibility and amino acid dissimilarity metric to produce the full EVEscape score. These additional metrics aim to weight the model towards detecting more immune-evasive mutations[36].

There are two mutability score variants[37], the direct coupling analysis (DCA) score and the Independent Site Model (IND) score. Both approaches use existing protein sequence alignments to calculate site-wise frequencies across the protein sequence. Much like the EVEscape fitness, the IND score is a likelihood ratio between the reference and mutated sequence. The DCA additionally incorporates coupling analysis to try to incorporate signals of mutational epistasis. This involves adding coupling terms between sites in order to approximate the epistatic interactions between them. We also tested other methods for computing grammaticality: masked grammaticality suggested by Allman et al.[39]; mutated grammaticality; and the relative grammaticality measure supplemented with the dissimilarity and accessibility as shown in Thadani et al.[36].

Experimental scores were used from three in vitro DMS studies, two of which were performed on the RBD while the other was performed on a full spike protein. "Escape", "Entry" and "Binding" were determined from a full spike DMS by Dadonaite et al.[9] and correspond to: immune escape from human sera, cell entry, and binding affinity to soluble ACE-2, respectively. "RBD Escape" was produced by Yisimayi et al.[40] using an RBD only DMS. The "Wuhan" and "Variant" binding and expression scores are from an RBD only DMS from Starr et al.[41] where

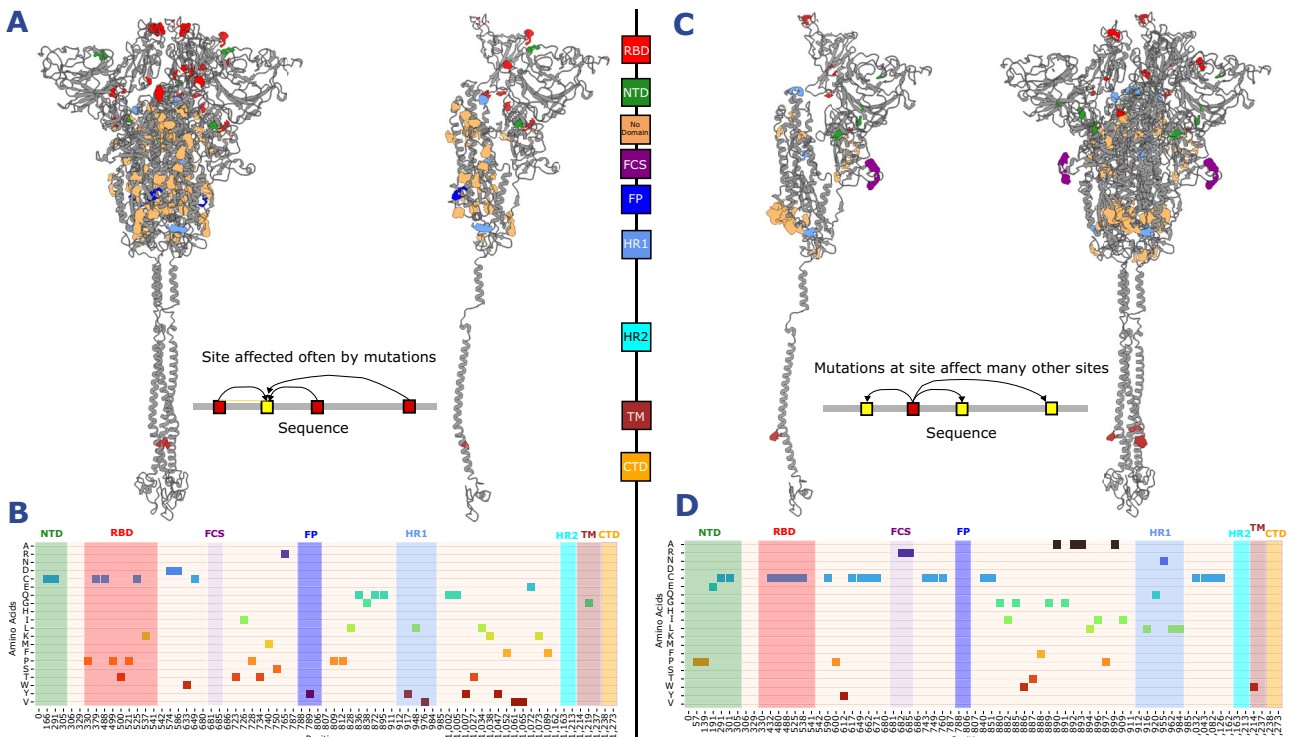

**Fig. 4 | ESM-2 identifies sites of constraint within Spike. A** The significantly changed logits for every mutation in the in silico DMS were identified, with positions that were repeatedly identified (called critical sites) as being affected by the in silico mutations counted. These were then mapped onto the spike protein structure and coloured on their domains. **B** Amino acids of each consistently affected reference residues from the DMS data. The NTD and RBD contain mostly

Prolines(P) and Cystines(C), while the rest of spike has a wider distribution of amino acids. **C** Sites that when mutated affect a significantly greater number of other positions (called impactful sites) in the sequence from the DMS data. **D** Reference residues for each of the impactful sites. An even larger proportion of impactful sites are cystines and are distributed throughout the protein, while arginines are enriched in the FCS.

binding to ACE-2 and expression of the protein were measured across five spikes. The Variant score is made from an average of both measurements across all five spikes.

Broadly, all metrics weakly correlate with experimental datasets (Fig. 5A). Semantic scores and grammaticality metrics are typically negatively correlated with each other. The semantic score exhibits a higher correlation on both "RBD Binding" datasets, while also achieving stronger correlations compared to grammaticality metrics on "RBD Expression" data. Grammaticalities tend to have higher correlations with Escape and full spike Entry and Binding data. However, the grammaticalities are typically only competitive with EVEscape and mutability scores when the accessibility and dissimilarity components of EVEscape were added to the relative grammaticality.

Regression correlation coefficients were comparable to the direct score correlations for most metrics (Fig. 5B). However, many were found to be insignificant upon multiple test correction. Full spike Entry and Binding were exceptions, with most predictions significantly yet weakly correlating. Logits and embeddings do significantly improve upon all single score metrics and were found to be significant in every experimental dataset. Correlation coefficients are also much higher, with all but full spike Escape and Binding achieving coefficients greater than 0.5. Embeddings and logits are dense representations, and much more descriptive than any single numeric metric. Non-linear regressions like SVR appear to extract the most from these representations, with linear regression typically achieving weaker correlations likely due to their inability to exploit non-linear relationships in these high-dimensional representations (Supplementary Fig. 11).

We also calculated several computational metrics that are commonly used for the analysis of protein sequences and estimation of

evolutionary constraints. These provide quantitative measures of substitution probability and structural characteristics at every site in the protein and are intended to be representative of the wide range of techniques currently used to analyse protein structural evolution. This allowed for comparison of the PLM metrics with existing metrics. The crystal structure of the spike protein was used to calculate "Accessibility", "B-Factor", "ΔΔG" and the environment-specific substitution table (ESST). Accessibility measures the antibody accessibility at every position. The B-Factor is the temperature factor derived from the protein crystallography experiment used to determine the 6VXX structure[42] and is a measure of the local fit of the structure to experimental data; it is often increased if there is static or dynamic disorder. Protein stability change (ΔΔG) was assessed with the FoldX software. The ESSTs were taken from Mizuguchi et al.[43] and provides a statistical estimate of substitution likelihood based on the observed frequency of substitutions at amino acids in similar local amino acid structural environments. We also calculated two sequence-based measures of substitution likelihoods. The position-specific scoring matrix (PSSM) was calculated using DeltaBlast and calculates the log likelihood of substitutions occurring based on their frequency in related sequences[42]. "Entropy" was used to provide an estimate of the variability present at a site.

Grammaticalities correlate better with computational metrics (Fig. 5A), particularly B-Factor and ESST. This indicates that grammaticalities (which align well with conservation) may also be a reasonable proxy for assessing protein flexibility. EVEscape and relative grammaticality combined with EVE components performed well on accessibility, but this is to be expected given that both use antibody accessibility as a component of the score. The semantic score performed better with PSSM and ESST but struggled to achieve high

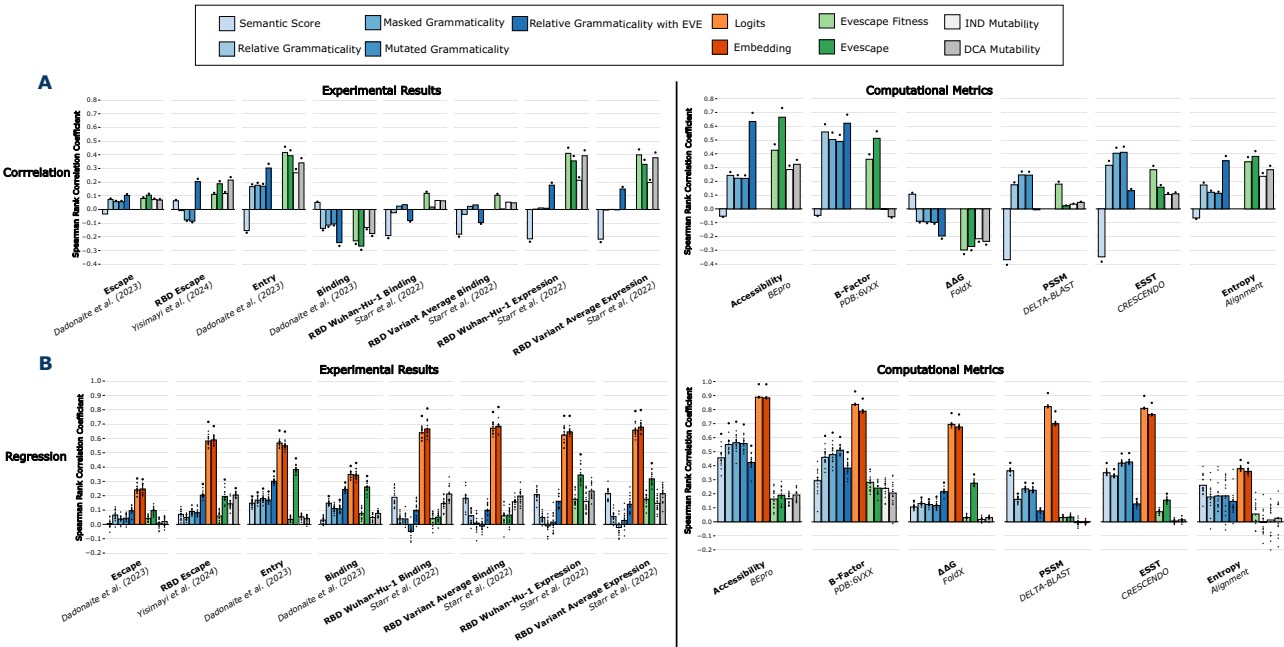

**Fig. 5 | ESM-2 derived PLM metrics do not directly relate to classical metrics.**
Comparison using (**A**) Spearman's Rank (two-sided) correlations (listed in Supplementary Data 1) between PLM and related metrics against traditional computational and experimental measures. Bars with an asterisk denotes the correlation was found to be significant after a Bonferroni correction (multiple test correction). **B** Spearman's Rank correlations (listed in Supplementary Data 1) where each

PLM (and related) metric was fitted using a support vector regression model with an RBF kernel to predict traditional and experimental measures. 5-fold cross validation was repeated on three randomised splits of the data. Bars represent the mean of the correlations. Bars with an asterisk denote the correlation was found to be significant after a Bonferroni correction.

correlations with other metrics. The relative grammaticality with EVE components was often competitive with EVEscape on most metrics, although typically EVEscape achieved slightly greater correlations more often.

Regressions improve correlations for most computational metrics (Fig. 5B), particularly for ESM-derived scores. Despite their accessibility components, EVEscape and relative grammaticality with EVE components both achieve worse correlations than the grammaticalities or the semantic score with SVR regression. Linear regression results are similar (Supplementary Fig. 11), and both EVEscape and mutability scores are found to be not significant. Grammaticality predictions are less correlated with B-Factor than the grammaticality scores however, are all found to be significant (Fig. 5B). EVEscape remains the most predictive score for ΔΔG but is only significantly predictive for one other computational metric ESST both with linear and SVR regressions. Embeddings and logits exhibit higher correlations with computational metrics when fitted with regression models, with all correlations above 0.65 except for Entropy, which remains challenging to predict. Larger representations like logits and embeddings therefore remain the most predictive but require fitting to data unlike single scores.

We also calculated a statistical distance metric, the Jensen Shannon distance (JSD), between these computational metrics and the semantic score and grammaticality at every site. Assessing the mean JSDs across different regions of the spike protein reveals a statistically significant difference between regions in S1 and S2 for the grammaticality score versus other metrics (Supplementary Fig. 6). This supports the conclusion that grammaticality carries implicit context-aware information about differences in evolutionary constraint between protein regions, as indicated by 2B; similar information not captured by existing, site-independent computational metrics. JSD between the semantic score and the other metrics does not show pronounced differences between S1 regions and S2 as consistently, although there were still statistically significant differences between some of the

protein regions, again demonstrating contextual structural awareness in the PLM embeddings.

We observe that the ESM-derived scores correlate weakly with the experimentally tested or computationally derived metrics (Fig. 5). However, when testing which mutations appeared in real data, i.e., SARS-CoV-2 pandemic sequences, we see that computational metrics are often equivalent to or better predictors than experimental measures, despite being significantly quicker to produce (Supplementary Fig. 10). To compare approaches equally, the DMS data were filtered to include measurements only available in all datasets. For comparison purposes the top decile of scores (as previously used by Thadani et al.[36]) were then selected for each method, and the number of mutations observed in the in VOC sequences (Supplementary Fig. 10A) and in the population (Supplementary Fig. 10B) were assessed.

The relative grammaticality with EVE annotations, EVEscape, and EVEscape Fitness were found to identify the most mutations out of the computational approaches (Supplementary Fig. 10A). ESM metrics identify many fewer of the VOC mutations without the additional accessibility and dissimilarity components used by EVEscape. This would suggest that most of the VOC mutations were driven by their accessibility to antibodies. We know that for several of the VOC defining mutations this is the case, with sites like 484 and 501 located within epitopes bound by multiple classes of antibody[24]. Both RBD binding measurements identify the greatest number of mutations for almost all measured variants with the top 10% of scores (Supplementary Fig. 10A). Full spike Binding and Escape were much less predictive and identified very few mutations in the RBD or in the rest of the spike. Full spike Entry identifies almost all of the mutations; however, this is mostly within the top 25% of scores. Given that experimental DMS was produced later in the pandemic from when a viable set of computational scores could have been created, computational methods remain competitive at identifying VOC-defining mutations.

Considering the broader set of mutations (Supplementary Fig. 10B), we see that ESM metrics outperform other DMS methods

when identifying mutations that occurred at higher frequencies (at least 100 times) in the pandemic data. The masked and mutated grammaticalities identify the greatest proportion of mutations in their top 10%. The masked score outperforms all experimental measures in the RBD and in the rest of spike. RBD experimental measures again identify more mutations than full spike DMS results. EVEscape fitness remains competitive with ESM-metrics, however EVEscape performs significantly worse. The relative grammaticality with EVE annotations also performs much worse here, although remains competitive with EVEscape. Mutations in this set don't all persist or become fixed unlike the VOC mutations. This indicates that many are likely to be close to neutral. This could explain why EVEscape and the relative grammaticality with EVE annotations perform poorly here, since these scores are weighted towards antibody accessible changes. Escape mutations often come at a cost, e.g., with many reducing binding affinity[44]. VOCs like Omicron include several mutations that individually seem deleterious but together efficiently evade host immunity[45].

While experimental results are considered the benchmark for comparisons, these computational metrics appear to be as predictive at identifying future mutations as the experimental measures while being significantly cheaper and quicker to produce. Experimental measures are more interpretable than ESM-derived scores since they assess a single property of a mutation. Sequence based language models derive their predictive signal from millions of protein sequences, spread across the tree of life. Hie et al.[6] show that their scores are associated with viral fitness and antigenicity, however, the alignments used to train their model contain changes that are heavily associated with immune evasion. This mirrors SARS-CoV-2's evolution during the pandemic, where most spike protein changes were the result of intense selective pressure to evade the host immune response and maintain viral fitness[1,24,44,45]. ESM-2 is trained on proteins that have a variety of selective pressures that are distinct from those faced by viral glycoproteins. This would suggest that an increase in the grammaticality or the semantic score is not simply a change in binding, escape, or any other single other protein property. Rather, they are a composite of these features, where associations to fitness or antigenicity can be drawn thanks to the over or under representations of similar sequences in the training data. Alignment entropy is predictive in much the same way, where the identification of changeable sites can help develop an understanding of protein properties, despite entropy not necessarily correlating with experimental results.

## Language models recapitulate evolutionary relationships between related sequences

Given that PLM approaches appear useful for interpreting single mutations within a protein sequence, we tested how ESM-2 would interpret differences between real SARS-CoV-2 sequences with multiple mutations. The earliest known SARS-CoV-2 sequence for each Pango lineage was extracted from the global data (retrieved from GISAID) and their spike protein sequences were embedded using ESM-2[5,7]. The evo-velocity package was then used to produce an evo-velocity UMAP for the sequence embeddings. Evo-velocity assigns a putative directionality between the embeddings that describes the flow of evolution through the UMAP embedding space. Firstly, evo-velocity[46], confirming these authors results, shows that PLM embeddings can distinguish between meaningfully different spike proteins (Fig. 6A, B). Secondly, it illustrates that embeddings can help to understand the evolutionary landscape of spike and the directionality of its evolution.

The evo-velocity accurately describes the evolution of SARS-CoV-2 moving from the earlier 'non-VOC' clusters into VOC clusters, consistent with their evolution during the pandemic (Fig. 6A, B). Omicron and Delta form distinct clusters while Gamma forms a concentrated cluster on the fringes of the non-VOC sequences. Beta and Alpha are less homogenous, and recombinants tend to fall with their parental

lineages (primarily Omicrons). Sarbecoviruses (which include bat and pangolin viruses related to SARS-CoV-2) form another distinct cluster close to the early SARS-CoV-2 sequences. These are also identified as the earliest sequences by the evo-velocity pseudotime algorithm (Fig. 6B and Supplementary Fig. 5). Evo-velocity first uses diffusion analysis to identify root and endpoint sequences before estimating the order of evolution using a pseudotime simulation (Fig. 6B and Supplementary Fig. 5). Pseudotime analysis achieved a significant ($p = 3.24e$-296) spearman's rank correlation of 0.86 against sampling time, confirming that the model has inferred the evolution of the sequences in the correct order. Evo-velocity recapitulates the topology of the representative sequence phylogeny in Fig. 6C, with early VOCs more closely related to non-VOC sequences than Omicrons and recombinants. This shows PLM embeddings capture meaningful representations that differentiate between distinct sequences. The congruence between the phylogenetic tree topology and the evo-velocity derived structure provides further confirmation that this method captures the evolutionary history of the spike protein.

PLM metrics also differentiate between distinct SARS-CoV-2 spike proteins. Using the semantic score or the relative grammaticality, a non-VOC, early VOC (Alpha, Beta, Gamma and Delta), and an Omicron cluster can be identified (Fig. 6, Supplementary Figs. 7 and 8). Unlike the UMAP (Fig. 6A and Supplementary Fig. 5) which is a dimensionality reduction and projection from high-dimensional space, grammaticality and the semantic score are much simpler to produce. Despite this, they still recapitulate much of the information shown in the UMAP. Evo-velocity, thus, captures differences between VOC sequences, recapitulates the topology of the nucleotide sequence phylogeny and reconstructs the direction of evolution accurately (Fig. 6). The PLM scores can effectively group lineages into their VOC categories (Fig. 6D, Supplementary Figs. 7 and 8).

## Detecting the distinct nature of the variants of concern on emergence

The emergence of SARS-CoV-2 and its subsequent variants of concern (VOCs) caused large waves of infections during the COVID-19 pandemic. Predicting their fitness advantage just from their initial sequences, before the viruses were in widespread circulation, has proven difficult. PLMs could potentially be used to meet this challenge of rapid characterisation of individual pathogen genomes as we have implemented for relative grammaticality and semantic scores (see Observable notebook:

https://observablehq.com/@cvr-bioinfo/from-a-single-sequencenature-communications).

The PLM metrics produced for each Pango lineage can be used to assess SARS-CoV-2 variants since they detect differences between sequences. We observed a characteristic "jump" in both semantic score and relative grammaticality between non-VOC, early VOCs and Omicron sequence clusters (Fig. 6D and Supplementary Fig. 8). Earlier VOCs required fewer changes to be successful since most of the human population was still naive to infection or vaccine-derived immunity. The Omicron lineage emerged during high levels of both vaccination and infection. It contained many more substitutions in the spike protein compared to previous variants, it changed its entry mechanism preference and managed to largely evade previous immunity which resulted in an extended vaccination regimen of three doses being recommended[47,48]. Unlike the other VOC groups, Omicron sequences initially decreased their semantic scores. The BA.2 variant had fewer mutations than the initial BA.1 variant, however later sequences that increased their number of mutations, continued to decrease their semantic scores. Clearly the semantic score is not a proxy for mutation count (Supplementary Fig. 9).

Once regular sequencing and surveillance are underway, a pipeline for analysing emerging sequences could identify outliers that may become future VOCs. Despite PLM scores distinguishing between

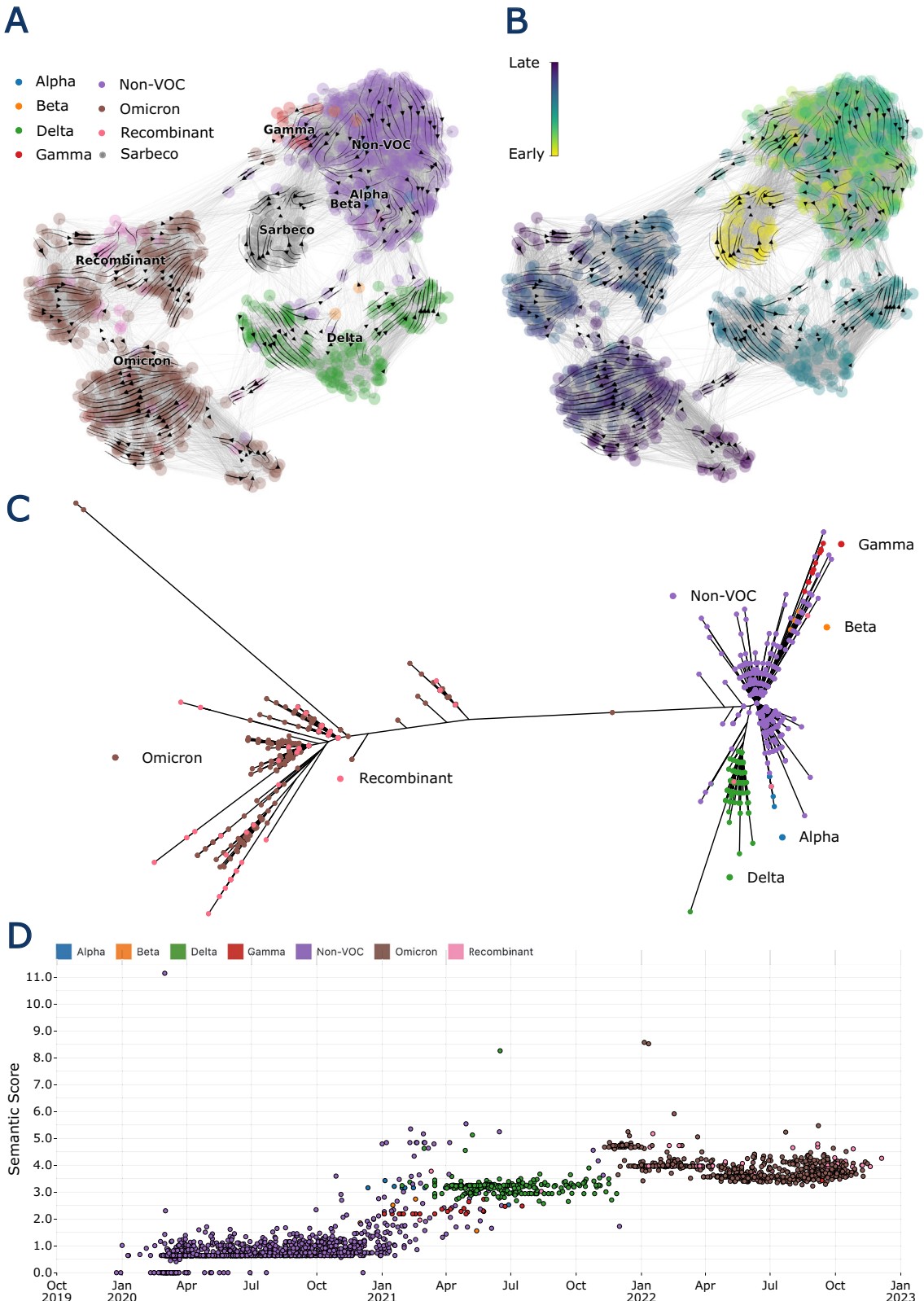

**Fig. 6 | ESM-2 recapitulates SARS-CoV-2s evolutionary trajectory. A** UMAP of initial spike sequence embeddings for SARS-CoV-2 Pango lineages and a selection of other known Sarbecovirus spike sequences. Each lineage is represented by one spike embedding. Points are coloured on VOC classification. Arrows represent the evo-velocity through the embedding space, which shows a "directionality" of evolution. **B** shows the sequences coloured by pseudotime inferred using sequence embedding probabilities to order sequences in time using an inferred root and an endpoint. **C** Shows an unrooted nucleotide phylogenetic tree of the spike sequences, coloured again by VOC. **D** shows the spike protein sequences plotted using their sample date and semantic score.

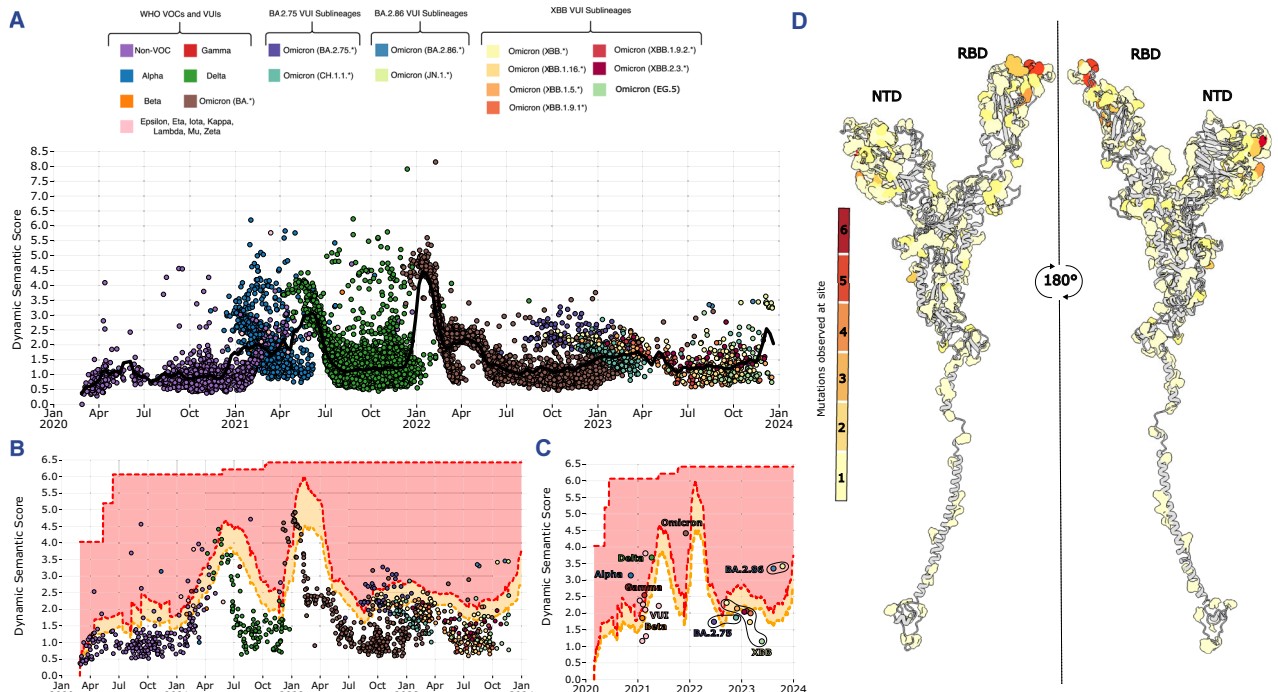

**Fig. 7 | Dynamic embedding metrics effective for identifying variants of concern. A** UK SARS-CoV-2 spike sequences through the pandemic. Each point represents a sequence cluster with 99.9% sequence similarity. Dynamic semantic scores were calculated for each sequence cluster, with the black line showing the mean sliding score. **B** Haplotypes are filtered on first occurrence of a new Pango lineage with standard deviations computed for each 3-month window. The orange warning area denotes sequences above 1 deviation from the mean, while the red warning area denotes >2 standard deviations. **C** Haplotypes filtered on the first appearance of a VOC/major variant, with deviation thresholds showing if they were detected on first appearance. **D** First haplotype sequence site mutation frequencies for sequences identified by warning thresholds mapped onto the spike monomer.

VOCs well, a larger score does not always mean a variant will be successful. This could be due to measuring against Wuhan-Hu-1, which is no longer circulating and increasingly irrelevant SARS-CoV-2 variant. To tackle this, we created a 'dynamic' embedding that represents the average sequence circulating at a given time. This means we can check whether sequences are both divergent from the SARS-CoV-2 reference or from what is currently or recently circulating. By looking at a UK subset of sequences, we can assess this approach thanks to the UKs high sequencing capacity paired with well-defined lineage waves. UK sequences were clustered to 99.9% similarity and a representative haplotype sequence for each cluster was embedded using the model.

Waves of semantic change take place between all major VOC waves in the UK (Alpha to Delta to BA.1 to BA.2) (Fig. 7A). The non-VOC to Alpha transition resulted in sequences deviating by a dynamic semantic score of >2, although this quickly increased to almost three after the emergence of Delta. Following BA.2, the semantic changes in Omicron have been more incremental, with a gradual increase up to March 2023, followed by a decrease until July 2023. There have been several smaller waves after Omicron's emergence, which suggests repeated dominance and replacement of consecutive distinct sublineages throughout this period. The haplotype embeddings also uncover the large diversity of semantic scores contained outside just the Pango representative sequences from Fig. 6D. Alpha and Delta have a large range of dynamic semantic scores, equalling some of the most divergent Omicron sequences despite pre-dating them by months (Fig. 7A). This is also true to a lesser extent for non-VOC sequences. What is clear is that as a new VOC began to take over, sequences diverge very quickly from the average circulating sequence embedding.

The dynamic semantic score can highlight outlier sequences, although these might be related to factors beyond sequence properties such as epidemiology which make variant prediction difficult for both experimental and computational methods. However, outlier

detection can filter out VOC-like sequences which are likely to be a subset of observed outliers. Using the first instances of each lineage (Fig. 7B), sequences above one and two standard deviations from the mean dynamic semantic score at each 3-month time point are classified as sequences of concern. We propose using these deviation thresholds as a sequence 'warning' indicators. When we filter for the first occurrences of known VOCs and important variants (Fig. 7C), we see that the majority (56%) are above one of the warning lines. These include all, at the time, WHO assigned VOC sequences (Alpha, Beta, Gamma, Delta and Omicron), with four of the five in the red 'warning' section. Three VUIs (Zeta, Iota and Mu) fall below the line, along with the BA.2.75 major variants and most of the XBB subvariants. VUIs were classified often due to their mutation composition or their early growth rates, yet many of these failed to maintain sustained spread due to competition from other variants. The Alpha variant was the dominant lineage in the UK when the VUI sequences arrived, which may explain their lack of spread in the UK, as well as their lack of risk classification. Alpha has a much larger dynamic semantic score than any of the early VOCs or VUIs and dominated in the UK until Delta's emergence.

The initial XBB lineage was detected as a sequence of concern by the outlier detector, however subvariants were almost exclusively below the thresholds for detection (except XBB.1.5). This is likely due to the high degree of similarity between the XBB spike and descendant lineages as most of the sub-lineages only differ by one or two sites. While BA.2.75 is not selected by its first sequences, several of its sub-lineages (including the second sub-lineage to emerge) are classified into the red warning zone (Fig. 7B). The heavily mutated BA.2.86 lineage acquired an L455S substitution[49] to produce JN.1, which subsequently became the dominant lineage circulating. Both BA.2.86 and JN.1 were identified by the outlier detection.

Using the haplotype sequences above the outlier warning lines, we counted the number of unique mutations in the spike protein that

occur within those sequences (Fig. 7D). The NTD and RBD contain the sites with the greatest quantity and diversity of changes at a single position. Of the mutations identified, 46% of the positions were also present in the first sequence of an identified major variant sequence, with 39% of the exact mutations present.

Emergent antigenically distinct SARS-CoV-2 variants are likely to cause reinfection[50]. Consequences of this are quickly becoming apparent through the necessity for updated vaccines, with recent variants evading previously neutralising antibodies[44,51]. We assert that the dynamic embedding approach is a viable horizon scanning method that does not simply count mutations but accurately models them and the shifting immune landscapes these lineages exist within. By identifying sequences of concern and contextualising their mutations within circulating viral sequences and host immune pressures, dynamic embeddings offer another useful and easy to implement technique in the event of a future outbreak.

In conclusion, we have shown that the PLM metrics grammaticality and semantic score[6] reveal characteristic properties of the SARS-CoV-2 spike protein sequence. Using just a single spike sequence we were able to determine the likely regions of variability in the spike protein, and identify regions where mutations were most likely to impact the structure and function of the protein. We describe how ESM-2 can map out evolutionary landscapes, identify epistatic effects and provide a method for horizon scanning of viral sequences of interest. Unlike other methods for predicting potential SARS-CoV-2 variant success, although representing improvements, for example, Ito et al.[52] and Thadani et al.[36], these need protein structures and multiple sequence alignments for training. In our implementation, the use of the pre-existing model ESM-2, negates the need for training and permits characterisation based on a single sequence. While there are clear areas for further improvement, this represents a paradigm shift for interpreting variation within protein sequences and could be applied to a novel pathogen when data is extremely sparse. A major strength of ESM-2 is its ability to learn broader evolutionary signals, which stems from its training data that spans the tree of life. While models trained from exclusively viral sequences may be produced in future, the sparsity of viral data included in training remains a major issue. Viral datasets currently contain only a small set of available and sufficiently distinct viral sequences. This is likely to limit the quality of any model inferences, however fine-tuning on such a set of sequences is more likely to yield improved performance. Models such as CovFit[52] and LucaVirus[53] have already demonstrated the power of fine-tuning general PLMs (LucaOne[54] and ESM-2[5]) on viral datasets, with more models likely to be released in future.

While ESM-2 is a good predictor of the computational and experimental metrics used to understand how mutations impact protein structures, these metrics do not correlate strongly with any one feature, indicating that PLM-derived scores are capturing something different about protein sequences. PLMs are now being used for DMS and variant effect prediction tasks across an impressive array of datasets, and often outcompete other methods[55]. They are helping improve the effectiveness of antibodies[56] and even generate new proteins[57]. Our results here affirm that these models are useful and can be applied to great effect to understand novel viral pathogens. PLMs, thus, offer an exciting glimpse into the future use of language modelling within biology, and we should continue to press on with understanding the possibilities as well as the limitations of what these models can do.

## Methods

### ESM-2 and in silico DMS

The ESM-2 model was used for most of the analysis in the paper. ESM-2 was trained on the September 2021 version of UniRef50 meaning that SARS-CoV-2 spike is present in the training data[5]. UniRef clusters are made using MMseqs2 to group together sequences with at least N%

identity overlap. UniRef90 is accordingly a set of clusters each represented by a sequence that has at least 90% identity to all other sequences in the cluster. UniRef50 is then a further clustering of the UniRef90 sequences where a sequence is selected for each cluster where the minimum identity is at least 50%. ESM-2 increases its diversity of sequences by using different UniRef90 sequences in certain minibatches during training. Since SARS-CoV-2 has a UniRef90 and UniRef50 cluster of which it is the representative sequence in both, it is very unlikely that anything other than the original SARS-CoV-2 Wuhan-Hu-1 spike protein would have been included in the training data. Further, the Omicron variant of the virus was not detected until November 2021[4,58] and at the time was the most divergent of the known circulating variants. The BA.1 lineage Omicron had an additional 39 differences (substitutions, insertions and deletions) relative to the Wuhan-Hu-1 spike protein, far below the number of differences required for the sequence to become its own cluster at either UniRef90 or UniRef50.

SARS-CoV-2 spike proteins were acquired by filtering the GISAID database (https://gisaid.org) for the earliest sequences from each Pango lineage with a fully intact spike protein sequence. The sequence was then embedded in the ESM-2 model to produce an embedding for the sequence. For the DMS data and for the embedding scores, the ESM-2 three billion parameter variant was used. The semantic score is equivalent to the L1 (Manhattan) distance between the embedding of the reference sequence (the spike protein from the original Wuhan-Hu-1 SARS-CoV-2 genome) and each of the Pango spike proteins. The grammaticality of a sequence is calculated as the product of the probabilities of each amino acid at each position in the spike protein. The probabilities come from a softmax of the last layer of the embedding and range between 0 and 1. However, many of the probabilities are small and for numerical stability the probabilities are represented in the log space. Relative grammaticality is the same as grammaticality, except the probability of a reference sequence is subtracted from the probability of the variant sequence so that the score is relative, in this case, to the SARS-CoV-2 reference sequence Wuhan-Hu-1. The sequence grammaticality represents the summed log-likelihoods of every reference position in the sequence, rather than just the mutated positions. For the DMS computations, a sequence was produced for every potential amino acid at every position in the spike protein. Each sequence was then embedded to calculate semantic scores and relative grammaticalities for every mutation.

### Extended grammaticalities

During the training of ESM, the model is asked to predict the amino acids present at mask tokens that account for a random 15% subset of each training sequence. Allman et al.[39] suggests that using the mask token is more appropriate for producing DMS scores, since this closely mirrors how the model is trained. This is different from Hie et al.[6], where the probabilities are not taken from the mask token at the site of interest, but from the reference token. We checked how the masked approach performs relative to the standard grammaticality from Hie et al.[6], while also investigating whether using the mutated sequence to calculate mutant probability was an appropriate measure, as this considers the rest of the sequence and the mutated site implicitly. Finally, we produced a grammaticality score that is weighted by the additional dissimilarity and accessibility components detailed in Thadani et al.[36] to see if ESM-metrics can also benefit from these additions. We included these alternative grammaticalities in the benchmarking (Fig. 5, Supplementary Fig. 10).

### Evo-velocity

We used the evo-velocity package[46] to embed the initial sequences using the ESM-2 650 M parameter variant, and then performed velocity analysis and spearman's rank for the sample dates against the pseudo-time. The smaller model was chosen primarily due to hardware

limitations of using the larger model, yet in several cases it has matched or even outperformed the larger model.

### Epistasis experiments

The epistasis experiments used a SARS-CoV-2 BA.1 spike protein sequence embeddings using the ESM-2 three billion parameter variant. Due to the EPA insertion, the logits for this position were subsequently removed to map logits to spike's three-dimensional protein structure. The BA.1 sequence contains several mutations relative to the Wuhan-Hu-1 SARS-CoV-2 reference spike sequence. Each of these mutations was reverted one by one back to the reference position, and the likelihood differences upon reversion were recorded. To eliminate noise, changes less than two standard deviations from the mean across all reverted mutations were removed and deemed not significant.

### Dynamic embeddings and horizon scanning

Dynamic embeddings were computed by first gathering UK GISAID data and clustering sequences into variants with 99.9% similarity. These haplotype variant clusters produced 11,272 sample date labelled haplotypes with a sequence returned for each cluster. Each haplotype embedding was measured against a mean of the embeddings from the prior three-month period using the L1 distance, i.e., the semantic score. For sequence grammaticalities, mean sequence grammaticalities were calculated in a similar way and differences measured.

### Assessing embeddings scores with known metrics

The language model's metrics were first assessed using a Spearman's rank against several known biological scores. Next, Support Vector Regression (SVR) was used as a simple model to fit the model scores as well as the embeddings and logits to the biologically relevant metrics. Models were fit to the data using five-fold cross validation, and a linear kernel for the SVR. Model results were reported as the average spearman's rank between the folds, with the error bars as +/-1 standard deviation from the mean. To assess statistical divergence between metrics at every site, scores were normalised to between 0 and 1, preserving rank order. The Jensen Shannon distance for each comparison was then calculated for every site independently using the philentropy (v.0.8.0) package in R. Amino acid positions used to define protein regions were as follows: S1 N-terminal (14-306), S1 RBD (331-528), S1 C-terminal (529-686), S2 (687-1273), all other positions were classed as indeterminate. Greater JSD implies greater divergence between two distributions. Significance was determined by a Mann–Whitney U test with Bonferroni correction.

### Selection analysis signals and entropy

Signals of ancestral evolutionary selection in the animal (bat and pangolin) sarbecovirus most closely related to SARS-CoV-2 referred to as the "nCoV" clade (see Lytras et al.[59]) were inferred on a set of 167 sarbecovirus genomes, accounting for recombination by inferring selection separately in each non-recombinant segment. These results are published in Martin et al.[60] and presented in more detail in the following Observable notebook: https://observablehq.com/@spond/ncos-evolution-nov-2021. Sites under negative selection were inferred using the FEL and sites under positive selection using MEME[61] by testing on internal branches of the nCoV clade. Sites denoted as conserved have the same amino-acid residue among all sarbecovirus sequences in the analysis. The variability of each site in the SARS-CoV-2 sequence was obtained from the entropy of the predicted distribution of credible evolutionary states.

### Antibody accessibility and substitution probabilities

Structure-based epitope score, referred to as "accessibility", which approximates antibody accessibility for each spike protein amino acid position, was calculated using BEpro software[62], see Harvey et al.[24]. Scores relating to substitution probabilities, namely, ESST probability,

Log PSSM and predicted ΔΔG, were obtained for every possible single amino acid substitution for the 6VXX SARS-CoV-2 spike structure (note that only values for Chain A are included in the results as data is generally identical across all three chains). ESST probability values were calculated using Environment Specific Substitution Table (ESST)[63] after local structural environments were calculated by JOY[43] Log Likelihood substitution values were calculated using Position Specific Scoring Matrices (PSSM) with the DELTA-BLAST[64] algorithm in BlastX. It should be noted that this is a sequence-based method, so residue numbering does not match the numbering in 6VXX and values are available for residues not described by the 6VXX PDB file. ΔΔG values were predicted by FoldX[65] software. FoldX uses empirical energy functions to predict the energetic effect of mutations to protein stability. The predicted ΔΔG quantifies the change in the free energy of unfolding between the wild-type and mutated structure. The 6VXX structure was first repaired with the RepairPDB function to fix residues with bad torsion angles, van der Walls' clashes or total energy. Substitutions were then performed using the repaired structures on all three chains simultaneously using the BuildModel function, giving the change in free energy of unfolding, with negative values implying stabilising mutations.

### Deep mutational scanning data

The receptor binding DMS data was taken from experimental DMS studies of the SARS-CoV-2 from Yisimayi et al. and Starr et al.[40,41]. The Wuhan-Hu-1 scores were taken as is, while the variant average score was calculated by averaging the scores for each position between each of the SARS-CoV-2 variant specific DMS results[41]. For the RBD mutational escape values we utilised the high-throughput mutation antibody escape profiling results presented in Yisimayi et al.[40]. This study used a panel of 1,350 monoclonal antibodies against all possible RBD substitutions. The backbone virus used was the SARS-CoV-2 BA.5 variant instead of Wuhan-Hu-1, however, this should still provide the most comprehensive dataset of unique substitutions' effect on antibody escape. Our mutational escape metric is the average of the raw antibody escape values for each substitution on each site of the RBD across all tested monoclonal antibodies. The full spike DMS data was taken from Dadonaite et al. and DMS scores ("Binding", "Escape" and "Entry") are from a BA.2 and XBB.1.5 spike backbone and averaged together to give the presented score[9].

### Reporting summary

Further information on research design is available in the Nature Portfolio Reporting Summary linked to this article.

## Data availability

The SARS-CoV-2 reference sequence Wuhan-Hu-1 (GenBank accession NC_045512.2) was used for the in silico DMS. The SARS-CoV-2 sequences for each Pango lineage are from GISAID (https://doi.org/10.55876/gis8.240620pm). The SARS-CoV-2 haplotype spike sequences from Fig. 7 are also from GISAID (https://doi.org/10.55876/gis8.240621ma). Seven of the Sarbecovirus sequences are from GISAID (https://doi.org/10.55876/gis8.241002yd) and 58 are from GenBank (accession numbers can be found in the Supplementary Data 2). The EVEscape benchmarking data (Supplementary Fig. 10) were collected from the GitHub repository (commit 8238e4f, https://github.com/OATML-Markslab/EVEscape/blob/main/results/summaries_with_scores/full_spike_evescape.csv, https://github.com/OATML-Markslab/EVEscape/blob/main/results/summaries_with_gisaid/spike_dist_one_scores_gisaid.csv and https://github.com/OATML-Markslab/EVEscape/blob/main/data/gisaid/single_mutant_count_by_month.csv). Data from DCA and IND mutability scores were collected from the GitHub repository (commit aeffe23, https://github.com/GiancarloCroce/DCA_SARS-CoV-2/blob/main/data/data_dca_proteome.csv). Supplementary Fig. 10A uses variants mentioned

in the spike_dist_one_scores_gisaid.csv file (https://github.com/OATML-Markslab/EVEscape/blob/main/results/summaries_with_gisaid/spike_dist_one_scores_gisaid.csv), but calculates the mutations from a representative set of sequences from GISAID (https://doi.org/10.55876/gis8.240620pm) so that all mutations (not just single nucleotide changes) could be identified. Supplementary Fig. 10B uses the data from single_mutant_count_by_month.csv (https://github.com/OATML-Markslab/EVEscape/blob/main/data/gisaid/single_mutant_count_by_month.csv), since this includes mutation frequency tracking for each month. This is restricted to just single nucleotide changes mutations. Epistasis benchmarking data from Innocenti et al.[30]. was downloaded from their supplementary table S1 (https://link.springer.com/article/10.1186/s13059-024-03355-y#Sec19). Two PDB structures were used in the analysis, the 6VXX Spike structure[42] (https://www.rcsb.org/structure/6VXX) and the Spike and ACE-2 simulated structure[66] 6vsb_1_1_6vw1 (https://charmm-gui.org/archive/covid19/6vsb_6vw1.pdb).

## Code availability

The code for the analysis as well as data can be found on GitHub (https://github.com/kieran12lamb/PLM_SARS-CoV-2). Code used to produce figures and the post-processed data can be found on Observable (https://observablehq.com/@cvr-bioinfo/from-a-singlesequence-nature-communications).

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

## Acknowledgements

We gratefully acknowledge all data contributors, i.e., the Authors and their Originating laboratories responsible for obtaining the specimens, and their Submitting laboratories for generating the genetic sequence and metadata and sharing via the GISAID Initiative, on which this research is based. The authors acknowledge funding from the UK Medical Research Council (MRC: MC_UU_12014/12 and MC_UU_00034/5 for D.L.R. and J.H.; MC_UU_00034/6 for D.L.R. and J.G.; MR/V01157X/1 and MR/Y002814/1 for D.L.R.) and a Doctoral Training Programme in Precision Medicine studentship (MR/N013166/1 for K.D.L.). D.L.R. and K.Y. acknowledge funding from the Wellcome Trust (220977/Z/20/Z). F.Y., D.L.R. and K.Y. acknowledge funding from the BBSRC (BB/V016067/1). D.L.R. also acknowledges support from the UK Research and Innovation (UKRI) to the G2P-UK consortium (MR/W005611/1) and G2P2 consortium (MR/Y004205), and the COVID-19 Genomics UK Consortium (COG-UK), which was supported by funding from the MRC, part of UKRI, the UK National Institute of Health and Care Research (MC_PC_19027) and Genome Research Limited, operating as the Wellcome Sanger Institute. K.Y. acknowledges support from Cancer Research UK (EDDPGM-Nov21\100001, DRCMDP-Nov23/100010 and core funding to the CRUK Scotland Institute (A31287)), Prostate Cancer UK (MA-TIA22-001) and EU Horizon 2020 (grant ID: 101016851). For the purpose of open access, the author has applied a Creative Commons Attribution (CC BY) licence to any Author Accepted Manuscript version arising.

## Author contributions

K.D.L. designed the experiments, collected datasets, wrote the code, contributed to the analysis of the experiments and prepared the manuscript. J.H., S.L., and J.C.H. contributed to the analysis of the experiments, collected datasets, and provided feedback on the experimental design. F.Y., O.K., S.C.L., and J.G. contributed to the analysis of the experiments. D.L.R. and K.Y. conceptualised the study, designed the experiments, edited the manuscript, and jointly supervised the research. All the authors discussed the results and commented on the manuscript.

## Competing interests

Ke Yuan is a co-founder and shareholder of TileBio Ltd.
