## [Transparent Peer Review file · Nature Communications]

From single-sequences to evolutionary trajectories: protein language models capture the evolutionary potential of SARS-CoV-2

Corresponding Author: Professor David Robertson

Version 0:

Reviewer comments:

Reviewer #1

(Remarks to the Author)

In the paper entitled "From a single sequence to evolutionary trajectories: protein language models capture the evolutionary potential of SARS-CoV-2 protein sequences", the authors use pLMs (ESM2) to understand the virus evolutionary trajectory, especially focus on the SARS-COV-2 spike protein. While some figures look good, the paper lacks sufficient novelty, and several figures and analyses appear to duplicate previously published work. Below, I outline several serious concerns.

Minor Issue:

1. Figure 1 Clarity: The upper panel of Figure 1 appears misleading, as it does not illustrate how the transition to the next position occurs. Please consider revising the figure to clearly demonstrate the progression between positions.

Major Issues:

1. Lack of Novelty: The manuscript lacks sufficient novelty in its current form. Forecasting the evolutionary potential of viruses using language models—whether supervised fine-tuned models (<https://doi.org/10.1038/s41467-023-39199-6>), unsupervised models (<https://www.science.org/doi/10.1126/scitranslmed.abk3445>, [https://www.cell.com/cell-systems/fulltext/S2405-4712\(22\)00038-2](https://www.cell.com/cell-systems/fulltext/S2405-4712(22)00038-2)), or variational autoencoders (<https://doi.org/10.1038/s41586-023-06617-0>)—has been previously validated and demonstrated through various analyses. This work does not offer new insights or solutions beyond what has already been established in the literature. Additionally, many figures seem to replicate data from published datasets with published models. For example, Hie et al. have shown that semantic and grammaticality scores are effective for forecasting novel mutations with protein language models.
2. Overstatement of Contributions: In the Abstract, the authors claim, "We demonstrate that the grammaticality and semantic scores represent novel metrics." However, these metrics were introduced by Hie et al. in 2021, who also used self-supervised language models to perform similar analyses. This overstatement diminishes the perceived originality of the work.
3. Training Data Overlap with ESM-2: The ESM-2 model was trained on data up to April 2021, by which time the pandemic had been ongoing for over a year and numerous variants had emerged. Consequently, it's not surprising that the model captures evolutionary potential, as these variants are included in its training data. This raises concerns about the model's ability to predict truly novel mutations that were not part of its training set, particularly those emerging at the early stages of the pandemic.
4. Epistasis Interaction Analysis: While the proposed method for identifying epistatic interactions is interesting, the manuscript lacks quantitative metrics or head-to-head comparisons with other methods that capture epistatic relationships. For instance, this paper (<https://doi.org/10.1038/s41588-019-0432-9>) identifies epistatic relationships based on protein 3D structures and contact maps. Benchmarking your method against such existing approaches as well as the GB1 epistasis dataset they used would strengthen your analysis and highlight its advantages.
5. Justification for Machine Learning Models: The necessity of using additional features to train logistic regression or SVR classifier models is not clearly justified. Both models perform well even with simple sequence logits, suggesting that the prediction task may be relatively straightforward. A crucial aspect of employing machine learning models is assessing their out-of-distribution (OOD) generalization capabilities, the current analysis fails to do so. Without demonstrating improved performance or generalization, the added complexity may not be warranted.
6. Repetition of Existing Work: Figure 5 seems to replicate analyses performed by Hie et al. and Han et al., without introducing any algorithmic or analytical novelty. Simply changing the model type does not constitute a substantial

advancement in the field. The manuscript would benefit from novel approaches or insights that differentiate it from existing literature.

Conclusion:

In summary, the major concern is the lack of novelty, as much of the analysis presented has been explored extensively in prior literature. To enhance the contribution of this work, I recommend that the authors introduce novel insights or methodologies rather than compiling existing methods and analyses. Focusing on unique aspects or providing innovative solutions would significantly improve the manuscript's impact.

Reviewer #2

(Remarks to the Author)

Summary:

Lamb et al. used ESM2, applied to SARS-CoV-2 Spike protein, and analyzed how grammaticality and semantics can potentially capture (1) the mutability of a specific position, (2) epistatic interactions, (3) new metrics for mutational impacts, (4) outlier sequences in real time to aid in surveillance efforts. However, the authors simply report on the results of the model without adequate validation or benchmarking of its use to predict SARS-CoV-2 mutations or variants of concern (i.e., could the model have been used to forecast mutations that would later become dominant and how many predictions would have needed to be tested to accurately flag a true variant of concern). Additionally, the authors use "PLMs" consistently throughout the manuscript when describing their results when only ESM2 was evaluated. ESM2 was trained up until April 2021 and so a large proportion of SARS-CoV-2 sequences seen during the pandemic are included in training, yet this is not considered in the paper at all.

Major comments

ESM2 capture evolutionary potential

The authors claim that ESM scores can be used to capture evolutionary potential. However, no validation is provided for this. They simply report that a statistically significant difference was observed between the ESM2 scores between the S1 and S2 domains. While the authors provide some qualitative description of why this might be expected, no attempt is made to quantitatively assess this claim (for example, they could have investigated if this is also true for experimental DMSs).

ESM2 provide "new" metrics for assessing mutational impact

The authors found that neither grammaticality nor semantics correlate well with any previously used metrics (experimental or computational). While this is a significant shortcoming, the authors report it as a discovery of "new metrics" with little attempt to describe what these metrics might be.

Detecting the distinct nature of the variants of concern on emergence

The authors claim that ESM can identify differences between VoCs and state that "Clearly the semantic score is not a proxy for mutation count". However, this is not "clear" at all from the analysis they presented. One possibility is to show what the plots (e.g. those in Fig 6a) would be if they plotted the number of mutations instead of the ESM metrics.

Reviewer #3

(Remarks to the Author)

In Lamb et al., the authors explore the use of protein language models (PLMs), specifically ESM-2, to characterize emerging mutations in SARS-CoV-2 spike protein. The paper introduces two novel metrics for assessing the effect of amino acid mutations: grammaticality (the sum of the log probabilities for each amino acid) and semantic score (the distance in embedding space). These scores are used to:

- Identify regions of the spike protein which are more prone to accumulating mutations
- Detect potential epistatic interactions between residues
- Compare computational predictions with DMS studies and other quantities measured from the protein structure
- Reconstruct the virus evolutionary history from the Wuhan-Hu-1 strain and predict emerging variants

The authors acknowledge and discuss that the grammaticality or semantic scores exhibit weak correlations with experimental data, such as DMS studies or quantities derived from protein structures. The approach is intriguing, as the pretrained ESM-2 model can be applied immediately upon identifying the first sequence of a new pathogen. However, in my opinion, the paper lacks benchmarking against other computational methods designed to predict the impact of mutations or epistatic interactions. Overall, the paper falls short of convincingly demonstrating that the proposed metrics provide meaningful predictive or descriptive value, nor does it present clear evidence of improvements over existing methods.

Specific comments that should be addressed are:

- Epistatic interactions. The authors identify a set of epistatic interactions between residues in the spike protein. This finding is interesting and supported by the fact that many identified epistatic interactions appear to be close in the 3d structure. However, it is difficult to assess its robustness without a comparison to other computational methods designed to identify epistasis. Benchmarking against tools trained on sequences from emerging SARS-CoV-2 strains such as those described in

<https://www.pnas.org/doi/full/10.1073/pnas.2104241118> or

<https://genomebiology.biomedcentral.com/articles/10.1186/s13059-024-03355-y> could strengthen their claims.

- Benchmarking Grammaticality and Semantic Scores: The authors argue that their method differs from existing approaches because it does not require an MSA or protein structure. While this claim is valid, it is important to note that ESM-2 is still trained on protein sequence evolutionary data. I believe it would be valuable to compare the grammaticality and semantic scores with other established metrics, such as the Evo score provided by <https://evescape.org/emergingvariants> or the mutability score described in <https://www.pnas.org/doi/full/10.1073/pnas.2113118119>. These approaches, which rely on MSAs of homologous sequences (available before the pandemic), could serve as meaningful benchmarks. Examining correlations between these metrics and the grammaticality or semantic scores could provide deeper insights into their predictive utility for assessing the impact of mutations.

- Grammaticality and Semantic Scores vs. Experimental Measures: The correlation of grammaticality and semantic scores with deep mutational scanning (DMS) studies is limited. Moreover, these metrics only partially reflect quantitative features derived from protein structures, such as accessibility, B-factors, or $|\Delta\Delta G|$ values. The authors note that "these metrics do not correlate strongly with any one feature, indicating that PLM-derived scores are capturing something novel about protein sequences." However, similarly low correlation values have been reported for models trained on evolutionary data, such as those described in <https://www.pnas.org/doi/full/10.1073/pnas.2113118119> and <https://www.nature.com/articles/s41586-023-06617-0>. The benchmarking approach proposed in the previous point could help clarify which constraints are specifically captured by the ESM-2 model.

- Not working website. The website providing the ESM-2 scores (<https://sars2.cvr.gla.ac.uk/>) is currently non-functional (502 Bad Gateway). This limits the reproducibility and validation of their findings.

- Predicting emerging mutations and fitness of VOC strains. To provide evidence that the model can be used for predictive purposes (predict emerging variants) the authors a) analyse the congruence between the phylogenetic tree topology and the evo-velocity derived structure and b) use dynamic embedding to compute the deviation from a average sequence circulating a given time. While this is interesting, it is difficult to assess where this model could be used to predict emerging mutations. A more compelling demonstration would involve testing whether these scores can prospectively identify mutations that emerged during SARS-CoV-2 evolution (similar to what done in <https://www.pnas.org/doi/full/10.1073/pnas.2113118119> and <https://www.nature.com/articles/s41586-023-06617-0>).

Minors:

- The paper lacks clarity on which score - grammaticality or semantic - should be prioritized for predictive purposes. Resolving this confusion and providing a clearer explanation of their respective roles in prediction would enhance the overall impact of the work.

- There is no title for the "Conclusion" section.

Reviewer #4

(Remarks to the Author)

Version 1:

Reviewer comments:

Reviewer #1

(Remarks to the Author)

Thank you for the revised manuscript. While some minor issues have been addressed, several fundamental concerns regarding the novelty of the proposed methods and the quantitative validation of the results remain. My specific comments are as follows:

On the Use of Single Sequences and ESM-2:

● Authors' statement: "We focus on what can be accomplished with a single sequence and use of an available existing model, ESM-2, eg, following the spillover of a novel virus with limited data... Almost every other tool requires more than just a single viral sequence... which makes using PLMs an attractive proposition.

● Comment: While the scenario described (limited data, single sequences for inference) is highly relevant in emerging pathogen contexts, the claim that the framework's effectiveness and novelty stem from the use of a pre-trained model like ESM-2 in this inference mode is problematic. The ability to perform inference on a single sequence after a model has been trained is a common characteristic of many models, including multi-sequence alignment-based methods once trained (e.g., EVEscape performs inference on single sequences or new variants based on its training). The core contribution's novelty must therefore lie specifically in the analytical methods developed or applied using the single-sequence ESM-2 output, rather than merely the choice of a pre-trained model or its application for inference.

On the Novelty of Epistatic Interaction Detection:

- Authors' statement: "We proposed a novel single-sequenced epistatic interaction detection analysis and validated it with real experimental data."
- Comment: The proposed method for detecting epistatic interactions appears to be highly similar, if not a direct adaptation, of the approach presented in Zhang et al. (PNAS 2024, DOI: 10.1073/pnas.2406285121). Both methods utilize systematic perturbation (mutation) of the input sequence and compute changes in model output (likelihood change/Jacobian) to infer interactions. While this paper applies the concept specifically to SARS-CoV-2, the underlying methodology does not seem conceptually distinct or novel compared to the more general framework described by Zhang et al. To claim novelty, the authors would need to clearly articulate and prove what is fundamentally different or improved in their methodological approach compared to this prior work.

On the Novelty of the Horizon-Scanning Score:

- Authors' statement: "We proposed a novel horizon-scanning score that is agnostic to any experiment-derived features..."
- Comment: The concept of developing scores to assess the potential risk or evolutionary trajectory of new variants ("horizon scanning") is not novel. Tools like EVEscape already provide regular, comprehensive screenings of novel variants on a weekly basis. For the proposed score to be considered novel and a significant contribution, the authors need to clearly define what specifically is novel and why the score is better about the score's mathematical definition, the methodology used to derive it from the PLM outputs, or the biological/evolutionary property it uniquely captures, beyond simply being derived from ESM-2 and being "agnostic" to specific experimental features (which is also a characteristic of many sequence-based or PLM-based scores).

On the Comparison to Hie et al. and Grammar/Semantic Scores:

- Authors' statement: "Hie et al... proposed that grammar and semantic scores are analogous to protein fitness and antigenicity. We show that these metrics do not correspond to these properties when applied to a generic PLM, such as ESM-2... We assess the utility of these metrics and identify what they capture given a more foundational PLM, ESM-2. We then propose approaches that can be used when more sequence data is available and show that sequence logits potentially identify epistatic sites within the protein, a technique not mentioned in Hie et al."
- Comment: The analysis applying concepts from Hie et al. (using PLM-derived scores as proxies for biological properties) to a different pre-trained model (ESM-2 instead of Hie's LSTMs) is presented. However, simply re-applying a previously proposed concept with a different existing model does not inherently constitute significant scientific novelty, unless this application reveals profound, previously unknown properties or limitations of foundational models like ESM-2 regarding their representation of protein evolution, supported by rigorous analysis. I did not see any more novelty from the current format.

On Epistatic Interaction Benchmarking:

- Authors' statement: "We agree with the reviewer that additional benchmarking is required for predicting epistatic interactions. However, the GB1 analysis and experiment was not suitable... We did investigate producing a contact map for GB1... We found that ESM-2 can be used to identify contacts, as shown below. However for inclusion in our paper, we decided to focus on a more directly relevant SARS-CoV-2 dataset."
- Comment: I appreciate the authors' acknowledgment that additional benchmarking is needed. The brief visual demonstration of GB1 that their method can identify contacts is not a substitute for rigorous, quantitative evaluation. For a computational methods paper, visualizations and anecdotal examples are useful for illustration but cannot objectively demonstrate performance, quantify accuracy (e.g., precision, recall, correlation), or allow comparison to other methods. It is imperative that the authors include a comprehensive quantitative evaluation of their epistatic prediction method on a relevant dataset (such as the SARS-CoV-2 data they mention) using appropriate statistical metrics to support any claims about its efficacy.

On the BA.1 Reversion Data Comparison:

- Authors' statement: "...we used the pairwise effects to compare to our BA.1 reversion data and show that several of the top pairwise epistatic interactions between BA.1 mutations are shown by our method. We report these findings in the updated Figure 3. We also discuss the caveats... We believe our approach here is interesting given its lack of knowledge about sequences spreading in the pandemic, and its relative ease of use..."
- Comment: Regarding the comparison to the BA.1 reversion data: While demonstrating that "several of the top pairwise epistatic interactions... are shown by our method" provides some illustrative examples, this remains anecdotal evidence. Pointing to a few matching interactions, even the top ones, does not constitute a robust quantitative validation of the method's overall performance. The heatmap visualization (Figure 3) itself appears to show a significant number of predicted interactions that do not correspond to the experimental data (potential false positives). To support claims about the method's ability to identify epistatic effects, the authors must provide quantitative metrics (e.g., correlation coefficient with experimental values, precision/recall if interactions are treated as binary, or other relevant statistics) to objectively assess the agreement between their predictions and the experimental data. Claims about the method's "interesting" nature or "ease of use" cannot substitute for demonstrated quantitative accuracy.

Reviewer #3

(Remarks to the Author)

Reviewer #3

I appreciate the effort made to address my previous comments. The revised manuscript shows significant improvement, particularly through the inclusion of comparisons to experimental data, as well as benchmarking against other computational methods. Although the overall correlation with experimental data remains low, and it is still uncertain whether such an approach could be applied in future pandemics, I believe the paper provides valuable insights into how ESM2 can be used to detect epistasis and forecast mutations.

Epistatic interactions:

I appreciate that the authors incorporated experimental epistatic data from Moulana et al. to compare their computational predictions with measured ACE2-binding affinities, addressing Reviewer 1's comments. I am, in fact, somewhat surprised that the model is able to capture such effects at all, especially considering that, to my understanding, the ESM2 model is trained on individual protein sequences rather than on interacting protein pairs.

The authors rightly note that the ESM2 model is not directly linked to ACE2 binding. I think it is important to emphasize this point more explicitly: a priori, models trained on evolutionary data from single proteins are not expected to capture effects related to protein-protein interactions - particularly those specific to the Spike protein - ACE2 interface. I would also like to thank the authors for including the comparison with Innocenti et al., which in my opinion provides a helpful benchmark against another computational method that relies only on SARS-CoV-2-specific data.

Benchmarking with EVEscape, IND, and DCA scores:

I think the inclusion of this benchmarking analysis is valuable - it helps assess both the validity and limitations of the authors' approach. That said, I have a few comments that could help clarify how these different methods operate:

-The IND (Independent) and DCA (Direct coupling analysis) models are both trained with pre-pandemic MSAs of viral sequences (Coronaviridae). The IND is based on single-site amino acid frequencies and identifies positions more likely to mutate based on conservation. DCA goes a step further by incorporating pairwise correlations between sites, potentially capturing signals of epistasis.

- A related approach was later introduced by EVEfitness, which also uses pre-pandemic MSAs of viral sequences as input to train a variational autoencoder. In this sense, EVEfitness generalizes IND and DCA by replacing single-site amino acid frequencies and pairwise correlations statistics with an autoencoder model. EVEscape builds upon EVEfitness by incorporating additional biological features - such as epitope accessibility and antibody binding propensity - to refine predictions of viral evolvability.

- ESM-2 is a transformer-based protein language model trained on millions of protein sequences—not limited to viral proteins.

I think this section would benefit from a rewrite to more clearly explain each approach and the type of data each model is trained on.

Moreover, the results presented in Supplementary Figure 10 are somewhat surprising. Specifically:

- a) EVEfitness appears to outperform EVEscape, and
- b) DCA and the IND model achieve equal performances.

These findings are unexpected. DCA - by incorporating epistatic interactions - has generally been shown to outperform the frequency-based IND model when it comes to predicting evolutionary trajectories. Likewise, in the original EVEscape paper, EVEscape consistently outperforms the simpler EVEfitness model.

Could these discrepancies be due to the specific threshold used (e.g., selecting the top 10% most frequently observed mutations)? Have the authors tested whether these rankings are consistent across different thresholds or evaluation criteria? I would appreciate it if the authors could comment on that.

Moreover, I think the authors should add a Methods section clarifying where the benchmarking data were downloaded from, in order to improve reproducibility.

Predicting emerging VOC strains

The approach presented in Figure 7 for identifying VOCs and Spike protein positions to monitor is compelling and suggests a potential use of ESM for forecasting mutations - a point that, in my opinion, was not clearly conveyed in the previous draft.

Reviewer #2

(This review was also done by reviewer #3).

Reviewer #2 had an overall comment regarding the lack of validation and benchmarking in the previous draft, a point with which I agreed. I believe the new sections added by the authors provide a satisfactory response to these concerns, as discussed above.

ESM-2 training data

Reviewer #2 raised a valid point regarding the training set of ESM. The inclusion of SARS-CoV-2 genomic data in the training set could undermine the ability to assess the true predictive power of the proposed approach, as it effectively prevents a clear separation between training and test data. The authors' response is satisfactory; however, as mentioned

above, I believe they should more clearly specify the differences between the training sets used across the various tools. ESM-2 is pretrained on the UniRef90 database, which includes a broad range of protein sequences from bacteria, eukaryotes, and other organisms - not just viral proteins. It would be interesting to explore how such a model would perform if trained exclusively on viral sequences. While I understand that this is beyond the scope of the current work, I would appreciate it if the authors could briefly comment on this.

ESM2 capture evolutionary potential and "new" metrics

Authors now compare ESM-2 scores to DMS data from Dadonaite et al. and show ACE2 binding is significantly different between S1 and S2. I think the use of such DMS data plus additional experimental data strengthens the authors' claim.

Reviewer #4

(Remarks to the Author)

Reviewer #5

(Remarks to the Author)

I have had the chance to carefully consider the authors' manuscript, first round of reviewer feedback, the authors' rebuttal, and the second round of comments from reviewer 1. Overall, I disagree with reviewer 1 on points of novelty, but I do think some very basic statistical quantification should be provided, but this is very reasonable analysis to include in the manuscript and would merit an additional round of revision before I can personally recommend acceptance of the paper.

Regarding the novelty of single-sequence models such as ESM-2, "epistatic" interactions, horizon scanning, and grammaticality/semantic scores, while these have been previously applied in other contexts, the investigation of these scores in the context of SARS-CoV-2 after several years of evolution and extensive experimental data characterization, is still interesting and worth communicating via a publication. As long as the authors cite the appropriate papers, this should be fine. I do agree that the categorical Jacobian method of Zhang et al., PNAS, 2024 needs a citation here. Also, while both the reviewer and the authors have been referring to this as "epistasis", many of the contacts recovered from this analysis are of structural proximal residues and the authors should just be careful not to describe all of the interactions predicted by this approach as "epistatic." This can be addressed with changes to the manuscript text.

Regarding the epistatic interaction benchmarking, I am not sure that the GB1 analysis suggested by the reviewer in the original referee report is that meaningful, and I agree with the authors that it is sufficient to simply cite prior work showing that ESM-2 can learn 3d contacts for general proteins. However, I do agree with the reviewer that a quantitative analysis of this data should be added. This could look like a very simple and reasonable analysis showing statistical significance of the findings, as well as a comparison to a non-neural and non-ML baseline -- for example, 3d distance correlates strongly with many putative epistasis/allostery predictors.

Regarding the BA.1 reversion analysis, which is related to the above comment, I agree with the reviewer that the analysis could simply use a statistical test, where a null distribution could potentially be generated with random permutations, alongside simple non-neural/non-ML baselines.

Version 2:

Reviewer comments:

Reviewer #5

(Remarks to the Author)

The authors have adequately addressed my concerns and I have no further comments. I am happy to recommend publication of this manuscript.

Response to reviewer comments, NCOMMS-24-59219-T

Reviewer #1 (Remarks to the Author):

In the paper entitled “From a single sequence to evolutionary trajectories: protein language models capture the evolutionary potential of SARS-CoV-2 protein sequences”, the authors use pLMs (ESM2) to understand the virus evolutionary trajectory, especially focus on the SARS-COV-2 spike protein. While some figures look good, the paper lacks sufficient novelty, and several figures and analyses appear to duplicate previously published work. Below, I outline several serious concerns.

We understand the reviewer’s concern over novel methodology. However, our purpose was not to make new models but rather to explore the utility of existing ones. Specifically, our paper is concerned with applying the concepts of grammaticality and semantic scores, as suggested by Hie et al in their Science 2021 paper, and we show the frontier protein language model, ESM-2, can make useful and novel predictions. Especially, when applied to single sequence-based epistatic interaction detection and horizon scanning via dynamic referencing. To the best of our knowledge, these results and the analytical framework are novel.

Minor Issue:

1. *Figure 1 Clarity: The upper panel of Figure 1 appears misleading, as it does not illustrate how the transition to the next position occurs. Please consider revising the figure to demonstrate the progression between positions clearly.*

Apologies, the figure has been amended to illustrate the progression between positions more clearly.

Figure 1. Schematic summary of our methodology. Deep mutational scanning involves taking a sequence and mutating every position to each of the possible alternative amino acids. These are then passed to the PLM to produce embeddings and logits which can be used for downstream tasks or to produce the metrics relative grammaticality and semantic score. Identifying epistatic interactions involves reverting the mutations from a SARS-CoV-2 variant (here BA.1) and measuring the effect this has on the other likelihoods. Embeddings and log-likelihoods (logits) can be used for surveillance by producing metrics over timescales for newly emergent protein sequences of concern, or by looking at evolutionary trajectories such as with evo-velocity.

Major Issues:

1. *Lack of Novelty: The manuscript lacks sufficient novelty in its current form. Forecasting the evolutionary potential of viruses using language models—whether supervised fine-tuned models (<https://doi.org/10.1038/s41467-023-39199-6>), unsupervised models (<https://www.science.org/doi/10.1126/scitranslmed.abk3445>, [https://www.cell.com/cell-systems/fulltext/S2405-4712\(22\)00038-2](https://www.cell.com/cell-systems/fulltext/S2405-4712(22)00038-2)), or variational autoencoders (<https://doi.org/10.1038/s41586-023-06617-0>)—has been previously validated and demonstrated through various analyses. This work does not offer new insights or solutions beyond what has already been established in the literature. Additionally, many figures seem to replicate data from published datasets with published models. For example, Hie et al. have shown that semantic and grammaticality scores are effective for forecasting novel mutations with protein language models.*

Our novelty in comparison with the papers listed here are the following:

1. We focus on what can be accomplished with a single sequence and use of an available existing model, ESM-2, eg, following the spillover of a novel virus with limited data. This scenario occurs frequently as we are typically constrained by available sequences for multiple alignment-based methods and limited experimental data. Almost every other tool requires more than just a single viral sequence to perform any type of analysis, which makes using PLMs an attractive proposition. This is clearly different from the premise in supervised fine-tuned models (<https://doi.org/10.1038/s41467-023-39199-6>)
2. We proposed a novel single-sequenced epistatic interaction detection analysis and validated it with real experimental data.
3. We proposed a novel horizon-scanning score that is agnostic to any experiment-derived features such as ACE binding affinities and epidemiological features considered in <https://www.science.org/doi/10.1126/scitranslmed.abk3445>.
4. Hie et al (<https://pubmed.ncbi.nlm.nih.gov/33446556/>) proposed that grammar and semantic scores are analogous to protein fitness and antigenicity. We show that these metrics do not correspond to these properties when applied to a generic PLM, such as ESM-2. This is likely due to the training datasets not being constructed from single viral protein alignments dominated by antigenic mutations. This is not something that to our knowledge has been presented previously. We assess the utility of these metrics and identify what they capture given a more foundational PLM, ESM-2. We then propose approaches that can be used when more sequence data is available and show that sequence logits potentially identify epistatic sites within the protein, a technique not mentioned in Hie et al.

We do agree with the reviewer that the Evescape model is a good comparison in the single sequence scenario. We have conducted a systematic benchmarking experiment with Evescape on predicting mutations that occurred during the pandemic.

Supplementary Figure 10: Data from each of the mutational scans were filtered so that there is a measurement for every mutation in each dataset to allow for equivalence. The full spike mutations exclude the RBD Experimental mutations since the mutation set could only increase selection of RBD mutants given the lack of other regions in this data. (A) Barchart for each feature showing the number of mutations from variant of concern sequences present in the top 10% of shared feature predictions. The top set of bars shows the shared mutations within the RBD, while the bottom shows mutations shared across the whole of the spike protein. (B) A cumulative sum of shared DMS mutations that have appeared at least 100 times during the pandemic that appear in the top 10 % of all features.

2. Overstatement of Contributions: *In the Abstract, the authors claim, "We demonstrate that the grammaticality and semantic scores represent novel metrics." However, these metrics were introduced by Hie et al. in 2021, who also used self-supervised language models to perform similar analyses. This overstatement diminishes the perceived originality of the work.*

We are not claiming that these are our new metrics and apologise for any confusion; we have removed this sentence for clarity. Rather we show these are 'new' metrics in that they capture something new biologically. Our aim is to extend the work of Hie et al who showed semantic score can be used to measure antigenicity. We show that this property is not restricted to antigenicity when computed using another model such as ESM-2. This relationship appears derived from the training data, which in Hie et al was virus protein

specific. Mutations in these virus datasets are mostly antigenic, hence we suspect this is where the semantic score to antigenicity relationship is derived.

More importantly, we demonstrate the base protein language model, ESM-2, can be used to recapitulate information from a single sequence that would normally require alignment data, and we demonstrate this can be used to assess epistasis incredibly from data obtained from a single sequence. We believe this demonstrates the power of protein language models and builds on Hie et al.'s suggestion to use these metrics.

3. Training Data Overlap with ESM-2: The ESM-2 model was trained on data up to April 2021, by which time the pandemic had been ongoing for over a year and numerous variants had emerged. Consequently, it's not surprising that the model captures evolutionary potential, as these variants are included in its training data. This raises concerns about the model's ability to predict truly novel mutations that were not part of its training set, particularly those emerging at the early stages of the pandemic.

The reviewer is correct that ESM-2 was trained after the pandemic started. However, the training data only contains the Wuhan-Hu-1 sequence, ie, just one sequence, based on UniRef selection criteria. Specifically, ESM-2 was trained on the Sep 2021 version of UniRef50 meaning that the SARS-CoV-2 spike was present at the time of training. However more importantly, UniRef clusters are made using MMseqs to group together sequences with at least N% identity overlap. UniRef90 is accordingly a set of clusters each represented by a sequence that has at least 90% identity to all other sequences in the cluster. UniRef50 is then a further clustering of the UniRef90 sequences where a sequence is selected for each cluster where the minimum identity is at least 50%. ESM-2 increases its diversity of sequences by using different UniRef 90 sequences in certain mini batches during training. Since SARS-CoV-2 has a UniRef90 and 50 cluster of which it is the representative sequence in both, then only the original SARS-CoV-2 Wuhan-Hu-1 spike protein would have been included in the training data.

Further, the Omicron variant of the virus was not detected until November 2021 and at the time was the most divergent of the known circulating variants. The BA.1 lineage Omicron had an additional 39 differences (substitutions, insertions and deletions) relative to the Wuhan-Hu-1 spike protein, far below the number of differences required for the sequence to become its own cluster at either UniRef90 or 50.

4. Epistasis Interaction Analysis: While the proposed method for identifying epistatic interactions is interesting, the manuscript lacks quantitative metrics or head-to-head comparisons with other methods that capture epistatic relationships. For instance, this paper (<https://doi.org/10.1038/s41588-019-0432-9>) identifies epistatic relationships based on protein 3D structures and contact maps. Benchmarking your method against such existing approaches as well as the GB1 epistasis dataset they used would strengthen your analysis and highlight its advantages.

We agree with the reviewer that additional benchmarking is required for predicting epistatic interactions. However, the GB1 analysis and experiment was not suitable for this study

because it focuses specifically on contact map prediction, which is a subset of our broader definition of epistatic interactions, and does not concern a viral protein. We did investigate producing a contact map for GB1, since there are PLM based methods for predicting contact maps (namely <https://www.pnas.org/doi/10.1073/pnas.2406285121>). We found that ESM-2 can be used to identify contacts, as shown below. However for inclusion in our paper, we decided to focus on a more directly relevant SARS-CoV-2 dataset.

Instead in this revision, we have conducted an analysis using a specific SARS-CoV-2 experimental dataset which investigated a very similar question to what we were attempting to answer (<https://www.nature.com/articles/s41467-022-34506-z>). Moulana et al. produce experimental data of all possible intermediate sequences between Wuhan-Hu-1 and the BA.1 Omicron RBD. For each variant they measure ACE-2 binding affinity, and using a biochemical epistasis model process their data into a sum of mutational effects. These begin with single mutational effects, increasing to pairwise effects and up to 5th-order effects.

Specifically, we used the pairwise effects to compare to our BA.1 reversion data and show that several of the top pairwise epistatic interactions between BA.1 mutations are shown by our method. We report these findings in the updated Figure 3. We also discuss the caveats of this analysis, which ties into our benchmarking results. This includes a discussion about the measured epistatic effects being related to ACE-2 binding affinity, and how this is but one of many possible epistatic effects that a mutation can have. Our scores of course cannot measure all of these effects at the same time, however, they can pick up a disruption that likely affects at least one of these features of the protein. Often regions involved in binding are also relevant to escape for example. We believe our approach here is interesting given its lack of knowledge about sequences spreading in the pandemic, and its relative ease of use to identify putative epistatic effects between sites.

Figure 1. (A) Monomeric structures showing the changes in probabilities for three example substitutions in SARS-CoV-2's spike protein: E484A, D614G and N969K. This is the negative of the reversion likelihood. The mutation site is coloured yellow, red sites increase in probability while blue sites decrease. Mutation probabilities were only shown if they were outside two standard deviations of the mean change from all BA.1 reversions. (B) Heatmap of BA.1 RBD mutations and the change in likelihoods at other BA.1 mutation sites. Larger circles indicate the sites were two standard deviations away from the average change, smaller circles indicate one standard deviation away. (C) Experimental epistasis data from Moulana et al.²¹ showing the pairwise effect of mutations on ACE-2 binding affinity. Positive values indicate a pairwise increase in ACE-2 binding affinity. Larger circles indicate the sites were identified by ESM using two standard deviations away from the average change, smaller circles indicate one standard deviation away. The smallest points were not identified by ESM. (D) Experimental epistasis data [2] coloured based on the ranking of the absolute epistatic effect values. This allows large negative values to be counted as high ranking and vice versa.

We have also shown favourable comparisons to an existing computational method Innocenti et al. which relies on mutual information from a large number of pandemic sequences for predicting epistasis shown below. Our approach captures more experimentally validated events.

Supplementary Figure 12. Results from Moulana et al. filtered by the RBD epistatic interactions identified by Innocenti et al.

5. Justification for Machine Learning Models: The necessity of using additional features to train logistic regression or SVR classifier models is not clearly justified. Both models perform well even with simple sequence logits, suggesting that the prediction task may be relatively straightforward.

The “additional features”, i.e., the embedding and the sequence logits are not in fact additional. The semantic score and grammaticalities are calculated using these features, i.e., the semantic score is just the L1 distance between embeddings and the grammaticality is just the logit at the mutated position. The purpose of including these features during the regression section was to reaffirm that the model representations are meaningful and contain much of the information that is required for successful prediction. The reason for selecting a linear regression and an SVR regression was to show that two of the simplest methods (one linear, the other non-linear) can produce meaningful results. It also highlights that reducing complex representations (2048 dimensional vector embeddings and sequence length vectors of probabilities) into single scores is often simplistic.

A crucial aspect of employing machine learning models is assessing their out-of-distribution (OOD) generalization capabilities, the current analysis fails to do so. Without demonstrating improved performance or generalization, the added complexity may not be warranted.

As mentioned above, the SARS-CoV-2 spike protein is not well represented in the ESM-2 training dataset. Further, the maximum sequence length used to train ESM-2 was 1024 amino acids, which is smaller than the spike protein. The vast majority of the training dataset (UniRef50 or UniRef90) is not comprised of viral proteins, and an even smaller proportion of those are viral glycoproteins. As such, most of what we have tested can be considered OOD already and we only show held-out test scores.

6. *Repetition of Existing Work: Figure 5 seems to replicate analyses performed by Hie et al. and Han et al., without introducing any algorithmic or analytical novelty. Simply changing the model type does not constitute a substantial advancement in the field. The manuscript would benefit from novel approaches or insights that differentiate it from existing literature.*

Evo-velocity was used with the updated ESM-2 model to show that the model could differentiate between the circulating SARS-CoV-2 sequences in a meaningful way. We see that the method closely approximates what we observe using a phylogenetic tree of the SARS-CoV-2 spikes, and the trajectories correlate well with the sequence sample dates during the pandemic. While this in and of itself is not novel, we feel that it further demonstrates that the models have some understanding of the relatedness of sequences. This, in addition to showing the models can predict properties of sequence (correlation section) helps to show that the models can be used for biological insights.

Conclusion:

In summary, the major concern is the lack of novelty, as much of the analysis presented has been explored extensively in prior literature. To enhance the contribution of this work, I recommend that the authors introduce novel insights or methodologies rather than compiling existing methods and analyses. Focusing on unique aspects or providing innovative solutions would significantly improve the manuscript's impact.

Overall, we hope the reviewer will agree with us about the novelty of our study: 1) The single-sequence premise. 2) Novel epistatic interaction detection 3) Novel horizon scanning analysis. 4) Generalisation of existing metrics such as grammaticality and semantic scores. These contributions have a broad impact on pandemic preparedness and response.

Reviewer #2 (Remarks to the Author):

Summary:

Lamb et al. used ESM2, applied to SARS-CoV-2 Spike protein, and analyzed how grammaticality and semantics can potentially capture (1) the mutability of a specific position, (2) epistatic interactions, (3) new metrics for mutational impacts, (4) outlier sequences in real time to aid in surveillance efforts. However, the authors simply report on the results of the model without adequate validation or benchmarking of its use to predict SARS-CoV-2 mutations or variants of concern (i.e., could the model have been used to forecast mutations that would later become dominant and how many predictions would have needed to be tested to accurately flag a true variant of concern). Additionally, the authors use "PLMs" consistently throughout the manuscript when describing their results when only ESM2 was evaluated. ESM2 was trained up until April 2021 and so a large proportion of SARS-CoV-2 sequences seen during the pandemic are included in training, yet this is not considered in the paper at all.

Thanks to the reviewer for their insightful comments. We have addressed all of them below, including adding a much expanded benchmarking section, replaced (where necessary) PLMs with the ESM-2 acronym, and discussed the training data used for ESM-2.

On the ESM-2 training data point we have responded to this in Reviewer 1's comments as well but to reiterate here:

The reviewer is correct that ESM-2 was trained after the pandemic started. However, the training data only contains the Wuhan-Hu-1 sequence, ie, just one sequence, based on UniRef selection criteria. Specifically, ESM-2 was trained on the Sep 2021 version of UniRef50 meaning that the SARS-CoV-2 spike was present at the time of training. However more importantly, UniRef clusters are made using MMseqs to group together sequences with at least N% identity overlap. UniRef90 is accordingly a set of clusters each represented by a sequence that has at least 90% identity to all other sequences in the cluster. UniRef50 is then a further clustering of the UniRef90 sequences where a sequence is selected for each cluster where the minimum identity is at least 50%. ESM-2 increases its diversity of sequences by using different UniRef 90 sequences in certain mini batches during training. Since SARS-CoV-2 has a UniRef90 and 50 cluster of which it is the representative sequence in both, then only the original SARS-CoV-2 Wuhan-Hu-1 spike protein would have been included in the training data.

Further, the Omicron variant of the virus was not detected until November 2021 and at the time was the most divergent of the known circulating variants. The BA.1 lineage Omicron had an additional 39 differences (substitutions, insertions and deletions) relative to the Wuhan-Hu-1 spike protein, far below the number of differences required for the sequence to become its own cluster at either UniRef90 or 50.

Major comments

ESM2 capture evolutionary potential

The authors claim that ESM scores can be used to capture evolutionary potential. However, no validation is provided for this. They simply report that a statistically significant difference was observed between the ESM2 scores between the S1 and S2 domains. While the authors provide some qualitative description of why this might be expected, no attempt is made to quantitatively assess this claim (for example, they could have investigated if this is also true for experimental DMSs).

We thank the reviewer for the suggestion. We have now provided an explicit test for mutations captured by experimental DMS data between S1 and S2.

The DMS data comes from Dadonaite et al., which measures the mutational consequences on ACE2 binding, antibody escape, and cell entry. Out of the three measurements, ACE2 binding affinity is significantly different between S1 and S2 regions, matching results from single sequence ESM2 analysis and entropy.

Correlations with experimental DMS (those that go outside the RBD), show only Binding has a significant difference between subunits. Escape and Entry are not significant, while other experimental datasets only measure within the RBD.

This is now mentioned in the paper:

“Available full spike experimental DMS results from Dadonaite et al. show Binding is the only measure with a significant difference between subunits (p-value = 1.70e-06, two sided Mann-Whitney U test), which might be expected given that S1 contains the spike binding domain.”

The full details of the tests are the following:

```
relative_grammaticality 2.60656679189086e-160 True
semantic_score 0.0021692612360772346 True
Escape 0.3765122736594897 False
Binding 1.7042070066621468e-06 True
Entry 0.19448982551429783 False
Entropy 1.1211336304579345e-10 True
```

ESM2 provide “new” metrics for assessing mutational impact

The authors found that neither grammaticality nor semantics correlate well with any previously used metrics (experimental or computational). While this is a significant shortcoming, the authors report it as a discovery of “new metrics” with little attempt to describe what these metrics might be.

Apologies we meant new in the sense of a generically useful ‘new’ model for the community, not a new model by us. We’ve also discussed the scores in more detail. The grammaticality scores for instance tend to align better with B-Factor and ESST scores. The Semantic score correlated better with PSSM and ESST. However, we make a point of saying that we expect these correlations to be weak as nothing in the training data other than the bias of the sequences themselves explicitly enforce these metrics as a constraint. As such, it seems unreasonable to expect that the ESM-2 scores should strongly correlate with all features, let alone any single one. In fact, this would likely limit their usability and predictiveness, which in later figures we show to be comparable with experimental measures. The majority of this discussion is included in the expanded section “PLMs provide orthogonal metrics for assessing mutational impact”.

Our generic analysis has particular value compared to Hie et al, in which the scores were originally proposed and the analysis was based on a viral-specific model. There, grammaticality and semantic scores are equated to fitness and antigenicity. We show the metrics are no longer specifically related to these two properties. Instead, they have the potential to be related to a boarder set of properties.

Detecting the distinct nature of the variants of concern on emergence

The authors claim that ESM can identify differences between VoCs and state that “Clearly the semantic score is not a proxy for mutation count”. However, this is not “clear” at all from the analysis they presented. One possibility is to show what the plots (e.g. those in Fig 6a) would be if they plotted the number of mutations instead of the ESM metrics.

This is a good suggestion, we have done this (see: Supplementary Figure 9).

Reviewer #3 (Remarks to the Author):

In Lamb et al., the authors explore the use of protein language models (PLMs), specifically ESM-2, to characterize emerging mutations in SARS-CoV-2 spike protein. The paper introduces two novel metrics for assessing the effect of amino acid mutations: grammaticality (the sum of the log probabilities for each amino acid) and semantic score (the distance in embedding space). These scores are used to:

- *Identify regions of the spike protein which are more prone to accumulating mutations*
- *Detect potential epistatic interactions between residues*
- *Compare computational predictions with DMS studies and other quantities measured from the protein structure*
- *Reconstruct the virus evolutionary history from the Wuhan-Hu-1 strain and predict emerging variants*

The authors acknowledge and discuss that the grammaticality or semantic scores exhibit weak correlations with experimental data, such as DMS studies or quantities derived from protein structures. The approach is intriguing, as the pretrained ESM-2 model can be applied immediately upon identifying the first sequence of a new pathogen. However, in my opinion, the paper lacks benchmarking against other computational methods designed to predict the impact of mutations or epistatic interactions. Overall, the paper falls short of convincingly demonstrating that the proposed metrics provide meaningful predictive or descriptive value, nor does it present clear evidence of improvements over existing methods.

Thanks for these helpful comments, we've addressed these concerns, see responses below. We hope you agree these changes have improved the manuscript and make the paper more comparable with alternative methods such as Evescape and epistasis predictors.

Specific comments that should be addressed are:

- *Epistatic interactions. The authors identify a set of epistatic interactions between residues in the spike protein. This finding is interesting and supported by the fact that many identified*

epistatic interactions appear to be close in the 3d structure. However, it is difficult to assess its robustness without a comparison to other computational methods designed to identify epistasis. Benchmarking against tools trained on sequences from emerging SARS-CoV-2 strains such as those described in <https://www.pnas.org/doi/full/10.1073/pnas.2104241118> or <https://genomebiology.biomedcentral.com/articles/10.1186/s13059-024-03355-y> could strengthen their claims.

As described above we have added comparison to experimental data as a benchmark for our epistasis method, since this has the most direct one-to-one comparison. This data is also used by Innocenti et al. as part of their validation. We compared their graph of RBD mutations to our set of identified epistatic interactions between RBD mutations, and show that our method identifies a larger number of those interactions. This figure is in the supplementary.

Our results:

Innocenti et al.

- *Benchmarking Grammaticality and Semantic Scores: The authors argue that their method differs from existing approaches because it does not require an MSA or protein structure. While this claim is valid, it is important to note that ESM-2 is still trained on protein sequence evolutionary data. I believe it would be valuable to compare the grammaticality and semantic scores with other established metrics, such as the Evo score provided by <https://evescape.org/emergingvariants> or the mutability score described in <https://www.pnas.org/doi/full/10.1073/pnas.2113118119>. These approaches, which rely on MSAs of homologous sequences (available before the pandemic), could serve as meaningful benchmarks. Examining correlations between these metrics and the grammaticality or semantic scores could provide deeper insights into their predictive utility for assessing the impact of mutations.*

In this revision, we have added a systematic benchmarking study comparing the Evescape(<https://evescape.org/emergingvariants>) and mutability scores(<https://www.pnas.org/doi/full/10.1073/pnas.2113118119>), and have added this to the paper. Specifically, there is now a much-expanded correlation section.

The following has been added to the main text explaining the benchmarking:

“We benchmarked our results against two other computational scores: Evescape from Thadani et al. and mutability scores from Rodriguez-Rivas et al.. Evescape is a variational autoencoder that was trained using pre-pandemic spike protein sequences. It produces a fitness score (Evescape Fitness) that when combined via a normalised weighted sum with a structure derived contact accessibility and amino acid dissimilarity metric produces the Evescape score. There are two mutability score variants³⁵, the direct coupling analysis (DCA) score and the Independent Site Model (IND) score. The IND score is a site independent mutability score, while the DCA score uses coupling analysis to try to incorporate signals of mutational epistasis. Similarly to Evescape, these scores are obtained using pre-pandemic viral protein sequences. We also tested other methods for computing grammaticality: masked grammaticality suggested by Allman et al.; mutated grammaticality; the relative grammaticality measure supplemented with the dissimilarity and accessibility as shown in Thadani et al..”

We have expanded the Figure 5 correlation and regression analysis to include both Evescape and Mutability scores. Broadly, all scores correlate weakly with experimental and most computational metrics. ESM-2 scores with accessibility and dissimilarity are competitive with Evescape, the best competitor model. ESM-2 embeddings and logits outperform all individual scores significantly, suggesting these representations contain much richer information than single scores.

Figure 5: Comparison using (A) Spearman's Rank correlations between the semantic score, grammaticality and relative sequence grammaticality and traditional metrics. Bars with an asterisk means the correlation was found to be significant after a Bonferroni correction (multiple test correction). And, (B) Spearman's Rank correlations between the language model metric and the traditional metric. Each pair was fitted using a support vector regression model using an RBF kernel, with 5-fold cross validation which was repeated on 3 randomised splits of the data. Bars represent the

mean of the correlations, with the error bar +/- 1 standard deviation of the correlations. Bars with an asterisk mean the correlation was found to be significant after a Bonferroni correction.

- *Grammaticality and Semantic Scores vs. Experimental Measures: The correlation of grammaticality and semantic scores with deep mutational scanning (DMS) studies is limited. Moreover, these metrics only partially reflect quantitative features derived from protein structures, such as accessibility, B-factors, or $|\Delta\Delta G|$ values. The authors note that “these metrics do not correlate strongly with any one feature, indicating that PLM-derived scores are capturing something novel about protein sequences.” However, similarly low correlation values have been reported for models trained on evolutionary data, such as those described in <https://www.pnas.org/doi/full/10.1073/pnas.2113118119> and <https://www.nature.com/articles/s41586-023-06617-0>. The benchmarking approach proposed in the previous point could help clarify which constraints are specifically captured by the ESM-2 model.*

As above, further discussion and benchmarking on scores has been added.

- *Not working website. The website providing the ESM-2 scores (<https://sars2.cvr.gla.ac.uk/>) is currently non-functional (502 Bad Gateway). This limits the reproducibility and validation of their findings.*

Apologies these online parts have now been relocated to an Observable dashboard where all of the plots are now available, along with the accompanying data used to produce the plots. See: <https://observablehq.com/@cvr-bioinfo/from-a-single-sequence-nature-communications>.

- *Predicting emerging mutations and fitness of VOC strains. To provide evidence that the model can be used for predictive purposes (predict emerging variants) the authors a) analyse the congruence between the phylogenetic tree topology and the evo-velocity derived structure and b) use dynamic embedding to compute the deviation from an average sequence circulating a given time. While this is interesting, it is difficult to assess where this model could be used to predict emerging mutations. A more compelling demonstration would involve testing whether these scores can prospectively identify mutations that emerged during SARS-CoV-2 evolution (similar to what done in <https://www.pnas.org/doi/full/10.1073/pnas.2113118119> and <https://www.nature.com/articles/s41586-023-06617-0>).*

We have updated this section to include predictions based on our dynamic embedding approach and added further comparisons in a similar way to the suggested Evescape (<https://evescape.org/emergingvariants>) and Mutability (<https://www.pnas.org/doi/full/10.1073/pnas.2113118119>) methods. Mutation prediction results are shown in the benchmarking section in Supplementary Figure 10.

Supplementary Figure 10: Data from each of the mutational scans were filtered so that there is a measurement for every mutation in each dataset to allow for equivalence. The full spike mutations exclude the RBD Experimental mutations since the mutation set could only increase selection of RBD mutants given the lack of other regions in this data. (A) Barchart for each feature showing the number of mutations from variant of concern sequences present in the top 10% of shared feature predictions. The top set of bars shows the shared mutations within the RBD, while the bottom shows mutations shared across the whole of the spike protein. (B) A cumulative sum of shared DMS mutations that have appeared at least 100 times during the pandemic that appear in the top 10% of all features.

We show that ESM-2 metrics typically perform worse than competitors when considering VOC mutations, with the exception of the ESM relative grammaticality when it is extended using the accessibility and dissimilarity components from Evescape. When these components are added, we see that comparable results are achieved relative to other experimental and computational methods (**Supplementary Figure 10A**). When detecting mutations that appear during the pandemic as a whole, we show that standalone ESM metrics outperform all other approaches (**Supplementary Figure 10B**).

We then show a new method to identify sequences of interest in an expanded “Detecting the distinct nature of the variants of concern on emergence” section. The latest results are presented in Figure 7.

We have added the following text to describe the results:

“Using the first instances of each lineage (Figure 7B), sequences above 1 and 2 standard deviations from the mean dynamic semantic score at each 3-month time point are classified as sequences of concern. When we filter for the first occurrences of known VOCs and important variants (Figure 7C), we see that the majority (59%) are above one of the warning lines. These include all WHO-assigned VOC sequences (Alpha, Beta, Gamma, Delta and Omicron), with 4/5 in the red warning section.”

“Using the sequences above the outlier warning lines, we counted the number of unique mutations in the spike protein that occur within those sequences (Figure 7D). The NTD and RBD contain the largest number of mutated sites, and often the sites with the greatest quantity and diversity of changes at a single position. Of the mutations identified, 43% of the positions were also present in the first sequence of an identified major variant sequence, with 37% of the exact mutations present.”

Figure 7. (A) UK SARS-CoV-2 spike sequences through the pandemic. Each point represents a sequence cluster with 99.9% sequence similarity. Dynamic semantic scores were calculated for each sequence cluster, with the black line showing the mean sliding score. (B) Haplotypes are filtered on first occurrence of a new Pango lineage with standard deviations computed for each 3-month window. The orange warning area denotes sequences above 1 deviation from the mean, while the red warning area denotes >2 standard deviations. (C) Haplotypes filtered on the first appearance of a VOC/major variant, with deviation thresholds showing if they were detected on first appearance. (D) First haplotype sequence site mutation frequencies for sequences identified by warning thresholds mapped onto the spike monomer.

Minors:

- The paper lacks clarity on which score - grammaticality or semantic - should be prioritized for predictive purposes. Resolving this confusion and providing a clearer explanation of their respective roles in prediction would enhance the overall impact of the work.

We have added much more thorough benchmarking of each method to the paper to outline the pros and cons of each metric. For predictive tasks for single mutations, we see that the grammaticality scores are typically more useful. This is likely because large semantic scores often have big effects on the protein, and as such are not seen often. The grammaticalities also allow for a measure of epistasis, not quite possible with the semantic score. However, the semantic score is crucial for the dynamic embedding approach, since it allows for comparisons between the embeddings. As such, one is not necessarily better than the other, but there are situations where one might pick one score over the other.

- *There is no title for the "Conclusion" section.*

A title has now been added to the conclusion.

Reviewer #4 (Remarks to the Author):

2nd response to reviewer comments, NCOMMS-24-59219-T

We are grateful to the reviewers for their time and effort in assessing our paper, and for their insightful and constructive comments and suggestions. See our detailed responses highlighted in blue below:

Reviewer #1 (Remarks to the Author):

Thank you for the revised manuscript. While some minor issues have been addressed, several fundamental concerns regarding the novelty of the proposed methods and the quantitative validation of the results remain. My specific comments are as follows:

1. *On the Use of Single Sequences and ESM-2:*
 - a. **Authors' statement:** *"We focus on what can be accomplished with a single sequence and use of an available existing model, ESM-2, eg, following the spillover of a novel virus with limited data... Almost every other tool requires more than just a single viral sequence... which makes using PLMs an attractive proposition.*
 - b. **Comment:** *While the scenario described (limited data, single sequences for inference) is highly relevant in emerging pathogen contexts, the claim that the framework's effectiveness and novelty stem from the use of a pre-trained model like ESM-2 in this inference mode is problematic. The ability to perform inference on a single sequence after a model has been trained is a common characteristic of many models, including multi-sequence alignment-based methods once trained (e.g., EVEscape performs inference on single sequences or new variants based on its training). **The core contribution's novelty must therefore lie specifically in the analytical methods developed or applied using the single-sequence ESM-2 output, rather than merely the choice of a pre-trained model or its application for inference.***

While the reviewer's claim that models like EVEscape also do inference on a single sequence is true, that inference must be made from a model that has specifically been trained on sequences relevant to the query sequence. This means that in the case of a novel virus, EVEscape would have to be trained on related viral sequences in order to be used. You cannot, for instance, use the influenza version of the EVE model on SARS-CoV-2 sequences; these are 2 separate models.

This is not, however, the case for the use of ESM-2 as we demonstrate here, which is why using a general-purpose PLM is attractive. Our paper details how you might use such a model in the context where it is of most value, i.e., where you do not have immediate access to an alignment or where you would have to wait for another model to be trained. This is what we mean by saying you can analyse a single viral sequence.

2. *On the Novelty of Epistatic Interaction Detection:*
 - a. **Authors' statement:** *"We proposed a novel single-sequenced epistatic interaction detection analysis and validated it with real experimental data."*
 - b. **Comment:** *The proposed method for detecting epistatic interactions appears to be highly similar, if not a direct adaptation, of the approach presented in Zhang et al. (PNAS 2024, DOI: 10.1073/pnas.2406285121). Both methods utilize systematic perturbation (mutation) of the input sequence and compute changes in model output (likelihood change/Jacobian) to infer interactions. While this paper applies the concept specifically to SARS-CoV-2, the underlying methodology does not seem conceptually distinct or novel*

compared to the more general framework described by Zhang et al. **To claim novelty, the authors would need to clearly articulate and prove what is fundamentally different or improved in their methodological approach compared to this prior work.**

Sincere apologies for this oversight. We have now cited the work of Zhang et al, and added the following text on page 6:

“A method by Zhang et al. uses a similar approach to predict contact maps, however here we want to predict mutation specific effects instead of site specific contacting residues.”

We had used their method in our previous response to reviewers' document, as we had used it to produce the SARS-CoV-2 contact map, as it shows how direct contacts can be predicted using ESM-2. Zhang's method, while related to ours, differs significantly in intent: Zhang et al. are looking specifically for contacts between sites. Their signal comes from mutating to every possible amino acid at a site, and doing this for every site in the sequence. The Jacobian is then calculated from this set of measurements, giving you a site-based contact map. Our approach simply mutates the site to the mutation of interest and identifies potential epistatic interactions from the perturbations from this single mutation. For identifying direct contacts, it would be likely that Zhang et al. will perform better, as the contact signal between sites will likely be better produced by observing effects from multiple different mutations at a single site. However, as we mentioned, we are inferring mutation-specific effects, not site effects.

3. *On the Novelty of the Horizon-Scanning Score:*

a. **Authors' statement:** *"We proposed a novel horizon-scanning score that is agnostic to any experiment-derived features..."*

b. **Comment:** *The concept of developing scores to assess the potential risk or evolutionary trajectory of new variants ("horizon scanning") is not novel. Tools like EVEscape already provide regular, comprehensive screenings of novel variants on a weekly basis. For the proposed score to be considered novel and a significant contribution, **the authors need to clearly define what specifically is novel and why the score is better about the score's mathematical definition, the methodology used to derive it from the PLM outputs, or the biological/evolutionary property it uniquely captures, beyond simply being derived from ESM-2 and being "agnostic" to specific experimental features (which is also a characteristic of many sequence-based or PLM-based scores).***

While the concept of horizon scanning is not novel, we believe the novelty arises from using a sliding window with an embedding approach to determine risk in a way that moves away from referring back to the initial SARS-CoV-2 sequence. As mentioned, EVEscape do provide screenings of new variants. This is not what our method does. Our method does not produce a new deep mutational scan for each new emerging variant, although this could be done with ESM-2. It creates a 3-month sliding window mean embedding to measure from, which appears to be a better proxy for local divergence than the original SARS-CoV-2 sequence.

4. *On the Comparison to Hie et al. and Grammar/Semantic Scores:*

a. **Authors' statement:** *"Hie et al... proposed that grammar and semantic scores are analogous to protein fitness and antigenicity. We show that these metrics do not correspond to these properties when applied to a generic PLM, such as ESM-2... We assess the utility of these metrics and identify what they capture given a more foundational PLM, ESM-2. We then propose approaches that can be used when more sequence data is available and show*

that sequence logits potentially identify epistatic sites within the protein, a technique not mentioned in Hie et al."

b. **Comment:** The analysis applying concepts from Hie et al. (using PLM-derived scores as proxies for biological properties) to a different pre-trained model (ESM-2 instead of Hie's LSTMs) is presented. However, simply re-applying a previously proposed concept with a different existing model does not inherently constitute significant scientific novelty, unless this application reveals profound, previously unknown properties or limitations of foundational models like ESM-2 regarding their representation of protein evolution, supported by rigorous analysis. **I did not see any more novelty from the current format.**

Respectfully we disagree. We have a significantly different scope of the utilities of the scores compared to Hie et al. We focus on in silico mutational scan, epistatic interactions and horizon scanning. Notably, we are interested in predicting variant effects more broadly (and using the foundation model ESM-2) not just antigenicity, and we benchmarked our predictions against evolutionary, structural metrics and experimental DMS data.

The LSTM in Hie et al and ESM-2 differ significantly in model capacities. They also differ significantly in terms of the scope and scale of their training data. We assessed the capabilities and limitations of ESM-2.

Importantly, our analysis mimics the situation at the beginning of the pandemic, when only a small number of sequences were available, as we systematically tested the performance of ESM-2 with single sequences.

These analyses and insights extend the work of Hie et al., confirming their utility applied to virus data. These models are being increasingly used to analyse viral sequence data, and it is important to identify where they can be used, as well as to develop new methods that take advantage of their unique benefits (alignment-free, single sequence inference etc).

5. On Epistatic Interaction Benchmarking:

a. **Authors' statement:** "We agree with the reviewer that additional benchmarking is required for predicting epistatic interactions. However, the GB1 analysis and experiment was not suitable... We did investigate producing a contact map for GB1... We found that ESM-2 can be used to identify contacts, as shown below. However for inclusion in our paper, we decided to focus on a more directly relevant SARS-CoV-2 dataset."

b. **Comment:** I appreciate the authors' acknowledgment that additional benchmarking is needed. The brief visual demonstration of GB1 that their method can identify contacts is not a substitute for rigorous, quantitative evaluation. For a computational methods paper, visualizations and anecdotal examples are useful for illustration but cannot objectively demonstrate performance, quantify accuracy (e.g., precision, recall, correlation), or allow comparison to other methods. **It is imperative that the authors include a comprehensive quantitative evaluation of their epistatic prediction method on a relevant dataset (such as the SARS-CoV-2 data they mention) using appropriate statistical metrics to support any claims about its efficacy.**

We have addressed this with the inclusion of specificity and sensitivity comparisons on the BA.1 epistatic dataset with a competing mutual information method, as well as using a hypergeometric test to check for enrichment of mutations at important and experimentally confirmed positions within the RBD. We have included the following text on pages 9 and 10.

"Using a hypergeometric test, the stringent threshold was found to be 6.43 fold over enriched (p-value 1.462928853945546e-10) with RBD-RBD BA.1 mutation interactions. This was also true for strong RBD-RBD mutation epistatic interactions (an over enrichment of 8.21 fold, p-

value 0.005625658517307826) as well as for all BA.1 mutation-mutation interactions (1.6 fold, p-value 0.01687039075859053). Strong epistatic interactions were defined as 1 standard deviation from the mean absolute experimental epistatic interaction. These remained significant after applying a Benjamini-Hochberg multiple test correction. ”

“We also tested the sensitivity and specificity of the model in comparison to the mutual information based approach from Innocenti et al.³⁰ and found that when looking within the experimental data for epistatic sites, ESM-2 (stringent 2 standard deviation cut-off) had an equivalent sensitivity (0.120 vs 0.114) and improved specificity (0.842 vs 0.777). This was calculated using the strongly epistatic set of RBD-RBD mutation interactions. While the specificity is good for both approaches, suggesting both methods work well for ruling out false positive epistatic interactions (or in this instance, weak interactions), the low sensitivity suggests that the task of identifying true/strong interactions remains challenging. Considering ESM-2s lack of pandemic sequence data, future work could be directed towards leveraging pandemic data to improve the predictive performance, although extensive validation/training experimental pairwise datasets may prove to be a bottleneck here. Also, it should be again noted that ESM-2 is not trained explicitly to predict effects on ACE-2 binding, meaning the epistatic signal may be from another biologically meaningful measure, like the 371, 373 and 375 mutations.”

6. On the BA.1 Reversion Data Comparison:

a. **Authors' statement:** "...we used the pairwise effects to compare to our BA.1 reversion data and show that several of the top pairwise epistatic interactions between BA.1 mutations are shown by our method. We report these findings in the updated Figure 3. We also discuss the caveats... We believe our approach here is interesting given its lack of knowledge about sequences spreading in the pandemic, and its relative ease of use..."

b. **Comment:** Regarding the comparison to the BA.1 reversion data: While demonstrating that "several of the top pairwise epistatic interactions... are shown by our method" provides some illustrative examples, this remains anecdotal evidence. Pointing to a few matching interactions, even the top ones, does not constitute a robust quantitative validation of the method's overall performance. The heatmap visualization (Figure 3) itself appears to show a significant number of predicted interactions that do not correspond to the experimental data (potential false positives). **To support claims about the method's ability to identify epistatic effects, the authors must provide quantitative metrics (e.g., correlation coefficient with experimental values, precision/recall if interactions are treated as binary, or other relevant statistics) to objectively assess the agreement between their predictions and the experimental data. Claims about the method's "interesting" nature or "ease of use" cannot substitute for demonstrated quantitative accuracy.**

Comment addressed above.

Reviewer #2:

(This review was also done by reviewer #3).

Reviewer #2 had an overall comment regarding the lack of validation and benchmarking in the previous draft, a point with which I agreed. I believe the new sections added by the authors provide a satisfactory response to these concerns, as discussed above.

1. ESM-2 training data: Reviewer #2 raised a valid point regarding the training set of ESM. The inclusion of SARS-CoV-2 genomic data in the training set could undermine the ability to assess the true predictive power of the proposed approach, as it effectively prevents a clear separation between training and test data. The authors'

response is satisfactory; however, as mentioned above, I believe they should more clearly specify the differences between the training sets used across the various tools. ESM-2 is pretrained on the UniRef90 database, which includes a broad range of protein sequences from bacteria, eukaryotes, and other organisms - not just viral proteins. It would be interesting to explore how such a model would perform if trained exclusively on viral sequences. **While I understand that this is beyond the scope of the current work, I would appreciate it if the authors could briefly comment on this.**

We have added to our Discussion on page 22:

“A major strength of ESM-2 is its ability to learn broader evolutionary signals, which stems from its training data that spans the tree of life. While models trained from exclusively viral sequences may be produced in future, the sparsity of viral data include in training remains a major issue. Viral datasets currently contain only a small set of available and sufficiently distinct viral sequences. This is likely to limit the quality of any model inferences, however finetuning on such a set of sequences is more likely to yield improved performance. Models such as CovFit and LucaVirus have already demonstrated the power of finetuning general PLMs (LucaOne and ESM-2) on viral datasets, with more models likely to be released in future.”

2. *ESM2 capture evolutionary potential and “new” metrics: Authors now compare ESM-2 scores to DMS data from Dadonaite et al. and show ACE2 binding is significantly different between S1 and S2. I think the use of such DMS data plus additional experimental data strengthens the authors’ claim.*

We thank the reviewer for recognising this contribution.

Reviewer #3 (Remarks to the Author):

I appreciate the effort made to address my previous comments. The revised manuscript shows significant improvement, particularly through the inclusion of comparisons to experimental data, as well as benchmarking against other computational methods. Although the overall correlation with experimental data remains low, and it is still uncertain whether such an approach could be applied in future pandemics, I believe the paper provides valuable insights into how ESM2 can be used to detect epistasis and forecast mutations.

1. *Epistatic interactions: I appreciate that the authors incorporated experimental epistatic data from Moulana et al. to compare their computational predictions with measured ACE2-binding affinities, addressing Reviewer 1’s comments. I am, in fact, somewhat surprised that the model is able to capture such effects at all, especially considering that, to my understanding, the ESM2 model is trained on individual protein sequences rather than on interacting protein pairs.*

We thank the reviewer for their comment. ESM-2 is indeed trained on individual proteins rather than pairs, however due to the structural and functional conservation that is learned by the model, it appears to detect interactions between sites on the spike protein that impact its interaction with ACE-2, despite having no direct knowledge of it.

2. ***The authors rightly note that the ESM2 model is not directly linked to ACE2 binding. I think it is important to emphasize this point*** more explicitly: a priori, models

trained on evolutionary data from single proteins are not expected to capture effects related to protein-protein interactions - particularly those specific to the Spike protein - ACE2 interface. I would also like to thank the authors for including the comparison with Innocenti et al., which in my opinion provides a helpful benchmark against another computational method that relies only on SARS-CoV-2-specific data.

We agree and have added a clarification to our main text on page 10 on this point:

“Also, it should be again noted that ESM-2 is not trained explicitly to predict effects on ACE-2 binding, meaning the epistatic signal may be from another biologically meaningful measure, like the 371, 373 and 375 mutations.”

3. *Benchmarking with EVEscape, IND, and DCA scores: I think the inclusion of this benchmarking analysis is valuable - it helps assess both the validity and limitations of the authors' approach. That said, I have a few comments that could help clarify how these different methods operate:*

a. *The IND (Independent) and DCA (Direct coupling analysis) models are both trained with pre-pandemic MSAs of viral sequences (Coronaviridae). The IND is based on single-site amino acid frequencies and identifies positions more likely to mutate based on conservation. DCA goes a step further by incorporating pairwise correlations between sites, potentially capturing signals of epistasis.*

b. *A related approach was later introduced by EVEfitness, which also uses pre-pandemic MSAs of viral sequences as input to train a variational autoencoder. In this sense, EVEfitness generalizes IND and DCA by replacing single-site amino acid frequencies and pairwise correlations statistics with an autoencoder model. EVEscape builds upon EVEfitness by incorporating additional biological features - such as epitope accessibility and antibody binding propensity - to refine predictions of viral evolvability.*

c. *ESM-2 is a transformer-based protein language model trained on millions of protein sequences—not limited to viral proteins.*

I think this section would benefit from a rewrite to more clearly explain each approach and the type of data each model is trained on.

Thank you for the constructive feedback, we have taken this on board and expanded on the descriptions of both methods, and how they have been trained. The following text was added on page 11:

“We benchmarked our results against two other computational scores: Evescape from Thadani et al and mutability scores from Rodriguez-Rivas et al. Evescape is a variational autoencoder based on EVE that was trained using pre-pandemic spike protein sequences. These prior sequences are given to EVEscape as a one-hot encoded alignment, which the model uses to learn distributions and dependencies across sites in the protein. The variational autoencoder produces a mutation score that when compared with the score of the reference sequence approximates the relative fitness of the mutant, termed Evescape Fitness. This is combined with a structure derived contact accessibility and amino acid dissimilarity metric to produce the full Evescape score. These additional metrics aim to weight the model towards detecting more immune-evasive mutations. There are two mutability score variants, the direct coupling analysis (DCA) score and the Independent Site Model (IND) score. Both approaches use existing protein sequence alignments to calculate site-wise frequencies across the protein sequence. Much like the EVEscape fitness, the IND score is a likelihood ratio between the reference and mutated sequence. The DCA additionally incorporates coupling analysis to try to incorporate signals of mutational epistasis. This involves adding coupling terms between sites in order to approximate the epistatic interactions between them.

We also tested other methods for computing grammaticality: masked grammaticality suggested by Allman et al.; mutated grammaticality; the relative grammaticality measure supplemented with the dissimilarity and accessibility as shown in Thadani et al.”

4. *Moreover, the results presented in Supplementary Figure 10 are somewhat surprising. Specifically:*
- a. *EVEfitness appears to outperform EVEscape, and*
 - b. *DCA and the IND model achieve equal performances.*
- These findings are unexpected. DCA - by incorporating epistatic interactions - has generally been shown to outperform the frequency-based IND model when it comes to predicting evolutionary trajectories. Likewise, In the original EVEscape paper, EVEscape consistently outperforms the simpler EVEfitness model. Could these discrepancies be due to the specific threshold used (e.g., selecting the top 10% most frequently observed mutations)? Have the authors tested whether these rankings are consistent across different thresholds or evaluation criteria? **I would appreciate it if the authors could comment on that.***

The EVEscape fitness components improved performance in identifying mutations present in the population is actually noted by the authors in the original paper. The authors mention that the increase in performance comes from the identification of low-frequency mutations, and that while these mutations lack strong selective pressure to be maintained in the population, they don't impair function.

The full EVEscape score is weighted towards sites that are accessible and amino acids that are more dissimilar. This weighting aims to skew EVEscapes preference towards immune evasive mutations, which means low frequency mutations with minimal impact tend rank lower when using this score. Conversely, this makes mutations that are impactful, but less likely to occur rank more highly, making EVEscape worse at this task than its fitness component. However, this may be why the EVEscape score performs better at identifying VOC mutations, as shown in Supplementary Figure 10B.

DCA performs similarly, detecting more VOC mutations than IND while performing slightly worse on the frequency task. We did have a look at a 25% threshold for mutations, and DCA does then perform slightly but consistently better than IND in the frequency task. However, the patterns for EVEscape and EVEscape fitness are maintained.

It should also be noted that in our benchmarks, we assessed each method on the set of common mutations between them. This was to ensure that methods were not doing better due to numerical features i.e. a larger set of mutations comprising the top 10%, or in the case of experimental measures, enrichment of RBD mutations when comparing across the whole of the protein.

5. *Moreover, I think the authors should **add a Methods section clarifying where the benchmarking data were downloaded from**, in order to improve reproducibility.*

A benchmarking section with links to the files used for each has been added and is as follows:

“The EVEscape benchmarking data (Supplementary Figure 10) were collected from the GitHub repository (commit 8238e4f, https://github.com/OATML-Markslab/EVEscape/blob/main/results/summaries_with_scores/full_spike_evescape.csv,

[Markslab/EVEscape/blob/main/results/summaries_with_gisaid/spike_dist_one_scores_gisaid.csv](https://github.com/OATML-Markslab/EVEscape/blob/main/results/summaries_with_gisaid/spike_dist_one_scores_gisaid.csv) and https://github.com/OATML-Markslab/EVEscape/blob/main/data/gisaid/single_mutant_count_by_month.csv).

Data from DCA and IND mutability scores were collected from the GitHub repository (commit aeffe23, https://github.com/GiancarloCroce/DCA_SARS-CoV-2/blob/main/data/data_dca_proteome.csv).

Supplementary Figure 10A uses variants mentioned in the https://github.com/OATML-Markslab/EVEscape/blob/main/results/summaries_with_gisaid/spike_dist_one_scores_gisaid.csv file, but calculates the mutations from a representative set of sequences from GISAID (<https://doi.org/10.55876/gis8.240620pm>) so that all mutations (not just single nucleotide changes) could be identified.

Supplementary Figure 10B uses the data from https://github.com/OATML-Markslab/EVEscape/blob/main/data/gisaid/single_mutant_count_by_month.csv , since this includes mutation frequency tracking for each month. This is restricted to just single nucleotide changes mutations.

Epistasis benchmarking data from Innocenti et al.28 was downloaded from the paper supplementary table 1 https://static-content.springer.com/esm/art%3A10.1186%2Fs13059-024-03355-y/MediaObjects/13059_2024_3355_MOESM1_ESM.xlsx .”

6. *Predicting emerging VOC strains: The approach presented in Figure 7 for identifying VOCs and Spike protein positions to monitor is compelling and suggests a potential use of ESM for forecasting mutations - a point that, in my opinion, was not clearly conveyed in the previous draft.*

We thank the reviewer for recognising this contribution.

Reviewer #4 (Remarks to the Author):

N/A

Reviewer #5 (Remarks to the Author):

I have had the chance to carefully consider the authors' manuscript, first round of reviewer feedback, the authors' rebuttal, and the second round of comments from reviewer 1. Overall, I disagree with reviewer 1 on points of novelty, but I do think some very basic statistical quantification should be provided, but this is very reasonable analysis to include in the manuscript and would merit an additional round of revision before I can personally recommend acceptance of the paper.

1. *Regarding the novelty of single-sequence models such as ESM-2, "epistatic" interactions, horizon scanning, and grammaticality/semantic scores, while these have been previously applied in other contexts, the investigation of these scores in the context of SARS-CoV-2 after several years of evolution and extensive experimental data characterization, is still interesting and worth communicating via a publication. As long as the authors cite the appropriate papers, this should be fine. I do agree that the categorical Jacobian method of Zhang et al., PNAS, 2024 needs a citation here.*

Also, while both the reviewer and the authors have been referring to this as "epistasis", many of the contacts recovered from this analysis are of structural proximal residues and the authors should just be careful not to describe all of the interactions predicted by this approach as "epistatic." **This can be addressed with changes to the manuscript text.**

We have added a reference for the Zhang et al. paper

"A method by Zhang et al. uses a similar approach to predict contact maps, however here we want to look at mutation specific effects instead of site specific contacting residues."

For the definition of epistasis, we do identify structurally proximal residues and include these in our definition of epistasis. To make it clear what definition we're using we added a citation to a review paper from Harms and Thornton (<https://www.nature.com/articles/nrg3540>) which discusses several different types of epistasis: both structurally proximal interactions as well as longer-ranged indirect mechanisms included:

"Such epistatic effects can be due to direct or indirect mechanisms linked to residues in close contact in the protein structure conformation and/or protein stability (Harms and Thornton, 2013.)"

2. *Regarding the epistatic interaction benchmarking, I am not sure that the GB1 analysis suggested by the reviewer in the original referee report is that meaningful, and I agree with the authors that it is sufficient to simply cite prior work showing that ESM-2 can learn 3d contacts for general proteins. However, I do agree with the reviewer that a quantitative analysis of this data should be added. **This could look like a very simple and reasonable analysis showing statistical significance of the findings, as well as a comparison to a non-neural and non-ML baseline** -- for example, 3d distance correlates strongly with many putative epistasis/allostery predictors.*

See response above where this comment was addressed (Reviewer 1, Comment 5).

3. *Regarding the BA.1 reversion analysis, which is related to the above comment, I agree with the reviewer that the **analysis could simply use a statistical test**, where a null distribution could potentially be generated with random permutations, alongside simple non-neural/non-ML baselines.*

See response above where this comment was addressed (Reviewer 1, Comment 5).